# Modeling the effects of alternative crop-livestock management scenarios on important ecosystem services for smallholder farming from a landscape perspective

Mirjam Pfeiffer[1], Munir P. Hoffmann[2], Simon Scheiter[1], William Nelson[3], Johannes Isselstein[4,7], Kingsley Ayisi[5], Jude J. Odhiambo[6], and Reimund Rötter[3,7]

[1]Senckenberg Biodiversity and Climate Research Centre (BiK-F), Senckenberganlage 25, 60438 Frankfurt am Main, Germany
[2]Agvolution GmbH, Phillip-Reis-Str. 2, 37075 Göttingen, Germany
[3]University of Göttingen, Tropical Plant Production and Agrosystems Modelling (TROPAGS), Grisebachstrasse 6, 37077 Göttingen, Germany
[4]University of Göttingen, Grassland Science, Göttingen, Germany
[5]Risk and Vulnerability Science Center, University of Limpopo, South Africa
[6]Department of Plant and Soil Sciences, Faculty of Science, Engineering and Agriculture, University of Venda, South Africa
[7]University of Göttingen, Center of Biodiversity and Sustainable Land Use (CBL), Büsgenweg 1, 37077 Göttingen, Germany

**Correspondence:** Mirjam Pfeiffer (mirjam.pfeiffer@web.de)

**Abstract.**

Smallholder farming systems in southern Africa are characterized by low-input management and integrated livestock and crop production. Low yields and dry-season feed shortages are common. To meet growing food demands, sustainable intensification (SI) of these systems is an important policy goal. While mixed crop-livestock farming may offer greater productivity, it implies trade-offs between feed supply, soil nutrient replenishment, soil carbon accumulation and other ecosystem functions and services (ESF and ESS). Such settings require a detailed systems understanding to assess the performance of prevalent management practices and identify potential SI strategies. Models can evaluate different management scenarios on extensive spatio-temporal scales and help identify suitable management strategies. Here, we linked the process-based models APSIM for cropland and aDGVM2 for rangeland to investigate the effects of (i) current management practices (minimum input crop-livestock agriculture), (ii) a SI scenario for crop production (with dry-season cropland grazing), and (iii) a scenario with separated rangeland and cropland management (livestock exclusion from cropland) in two representative villages of the Limpopo province, South Africa, for the period 2000-2010. We focused on the following ESF and ESS provided by cropland and rangeland: yield and feed provision, soil carbon storage, cropland LAI, and soil water. Village surveys informed the models of farming practices, livelihood conditions, and environmental circumstances. We found that modest SI measures (small fertilizer quantities, weeding, crop rotation) led to moderate yield increases between factor 1.2 and 1.6 and reduced soil carbon loss, but sometimes caused increased growing season water limitation effects. SI effects therefore strongly varied between years. Dry-season crop residue grazing reduced feed deficits by approx. factor 2 compared to the rangeland-only scenario, but could not fully compensate the deficits during the dry-to-wet season transition. We expect that targeted deficit irrigation or measures to improve water retention and soil water holding capacity may enhance SI efforts. Off-field residue feeding dur-

ing the dry-to-wet season transition could further reduce feed deficits and reduce rangeland grazing pressure during the early growing season. We argue that integrative modeling frameworks are needed to evaluate landscape-level interactions between ecosystem components, evaluate the climate resilience of landscape-level ecosystem services, and identify effective mitigation and adaptation strategies.

*Key words:* smallholder farming, mixed crop-livestock farming, Southern Africa, ASPIM, aDGVM2, vegetation modeling, dry-season feed gap, sustainable intensification

Dedicated to the memory of our esteemed colleague Marian Koch

## 1 Introduction

Smallholder farms occupy approximately 62% of Africa's farmland area. Family-driven labor generates household income and food security (FAO, 2014) and heavily relies on natural ecosystem functions and services (ESF and ESS), including the provision of yield and biomass as services provided by nature that benefit farmers and their livelihoods (Costanza et al., 1997; Millennium Ecosystem Assessment, 2005). Low income and education levels, poor access to markets and credits, lack of technology, and strong dependence on external support from governments and NGOs in form of safety nets, advisory support, off-farm income such as pensions, or donations, are challenges that endanger food security, welfare, and well-being of smallholders (FAO, 2015). These obstacles impede the capacity to mitigate and adapt to climate change (Harvey et al., 2013). Livestock husbandry and subsistence cropping characterize smallholder farming in Africa (Thornton and Herrero, 2015; Descheemaeker et al., 2016). Livestock provide milk, meat, and leather, convey prestige, and contribute to economic diversification. Livestock often feed on crop residues during the dry period, which allows rangeland resting, accelerates nutrient recycling, and links both land-use types. Although integrated crop-livestock farms may offer greater farming efficiency and sustainability (Sumberg, 2003; Herrero et al., 2009; Tarawali et al., 2011) they often lead to various trade-offs (Herrero et al., 2009; Erenstein, 2002). Harvest residues as fodder reduce feed gaps whereas leaving them on-field allows for nutrient replenishment and enhances soil fertility (Castellanos-Navarrete et al., 2014; Valbuena et al., 2012). However, freshly excreted nutrients from livestock are not used during dry-season fallow and considerable nutrient losses can occur (Hack-ten Broeke et al., 1996).

Climatic variability creates additional challenges. Southern Africa experiences high inter-annual rainfall variability due to the El Niño–Southern Oscillation phenomenon (Conway et al., 2015). Increased stocking density during high-rainfall years results in forage shortages in subsequent dry years. Over-stocking without feed supplementation then results in severe pressure on the drought-afflicted vegetation and leads to animal malnutrition, increased livestock mortality, economic losses, and threats

to ESS provided by rangelands (Müller et al., 2015). Overgrazing may lead to rangeland degradation, species loss, reduced carbon transfer to soils, and habitat deterioration. Higher bare ground fractions increase run-off, soil erosion, and evaporation.

Focusing on the multi-functionality and complex interactions of mixed cropland-rangeland systems is necessary to evaluate the performance of prevalent cropping and livestock husbandry practices and identify possible site-specific sustainable intensification strategies (Giller et al., 2006; Rusinamhodzi et al., 2011). Observational assessments of different management options requires numerous field experiments, which are often limited by time and resource constraints. In contrast, models can systematically explore different management scenarios on various spatiotemporal scales and help identify suitable management strategies once they have been evaluated satisfactorily (Kersebaum et al., 2015). Crop simulation models (CSMs) simulate the effects of different management strategies on biomass, yield, water use and nutrient uptake for numerous combinations of genotype and environment interactions (see, e.g., Rötter and Van Keulen, 1997; Whitbread et al., 2010). Livestock models that simulate animal productivity (meat, milk) depending on age, gender, and status (e.g., juvenile, pregnant, or lactating, see van de Ven et al., 2003) can integrate the output of such CSMs. For example, Descheemaeker et al. (2018) used the crop model AP-SIM (Holzworth et al., 2014) and the livestock model Livsim (Rufino et al., 2015) within the AgMIP framework (Rosenzweig et al., 2013) to investigate climate change effects on forage availability and livestock and crop productivity. Dynamic vegetation models (DVMs) simulate natural vegetation dynamics, carbon sequestration, energy and water fluxes, and biomass production in response to environmental drivers and disturbances. Recent developments increasingly focused on anthropogenic influences and aim to include management. For example, the aDGVM, a DVM developed for African savanna ecosystems (Scheiter and Higgins, 2009), has been expanded to simulate fuelwood harvesting and grazing to determine how climate change influences the economic value of ESS in southern African rangelands (Scheiter et al., 2019). New routines in the trait-based aDGVM2 allow for an improved representation of grass-layer diversity and simulation of selective grazing (Pfeiffer et al., 2019). Currently, livestock in aDGVM2 are not represented as an interactive agent, i.e., the model simulates livestock impact on vegetation but cannot simulate herd-related aspects such as decision-based animal movement, growth, metabolism, reproduction, or nutrition status in the manner of an agent-based model such as RaMDry (Fust and Schlecht, 2018).

While assessments of ESF in crop-livestock systems are crucial for smallholder farming communities in sub-Saharan Africa (Descheemaeker et al., 2016; Waha et al., 2018), CSMs and DVMs commonly consider cropland and rangeland independently. While such applications improve the individual system understanding, only a coupled framework that combines CSMs and DVMs can capture landscape-scale interactions. In this study, we linked APSIM with aDGVM2, two models that are both well-tested for southern Africa (Hoffmann et al., 2018a, b, 2020; Pfeiffer et al., 2019), to address the following research questions for a landscape-scale study including two villages in South Africa's Limpopo province:

1. Do village-specific sustainable intensification measures improve ecosystem services and functions compared to status quo land use practice, and which functions and services are affected?

2. Does sustainable intensification lead to stronger positive effects on environmentally constrained sites compared to environmentally more favorable sites?

3. Do feed gaps occur and are there village-specific differences?

4. Can joint management of cropland and rangeland reduce or eliminate feed gaps?

    5. Can integrated modeling of cropland and rangeland identify management strategies that result in a sustained provision of landscape-level ecosystem functions and services?

Simulations of cropland and rangeland dynamics for the two villages were coupled offline to test the impacts of different management scenarios on those landscape-scale ESS that are a) particularly relevant for smallholder farmers in the region and b) can be quantified by the coupled modeling framework. These include crop yield, provision of biomass for animal consumption from rangeland and cropland, carbon input to cropland soils and cropland soil organic carbon (SOC) formation and sequestration, soil water availability for crops, and LAI as a measure to describe soil cover and, implicitly, protection against erosion and potential photosynthetic capacity at stand level. Explicit evaluation of additional ESF such as run-off, evapotranspiration or species diversity was not part of the current study, although we acknowledge them as relevant. We focused on the following scenarios: the current status quo (minimum input integrated crop and livestock agriculture), a minimum sustainable intensification scenario, and separated vs. combined cropland-rangeland management. The APSIM model delivered results on yield, biomass productivity, carbon sequestration and water use. We then used this output to compare feed demand with simulated harvest residues to determine whether cropland can cover the feed demand during dry seasons. The aDGVM2 simulated rangeland vegetation dynamics and biomass consumption during periods when livestock had no cropland access. The aDGVM2 output showed whether rangeland could satisfy animal demand during those periods. Moreover, seasonal and interannual high-risk times for feed deficits were determined. Such an integrative analysis combines the strengths of crop and rangeland models and is the first attempt in this form.

## 2   Material and Methods

### 2.1   Study region

South Africa's Limpopo province has a high share of smallholder farmers who often experience food insecurity. Maize is a staple crop, accompanied by legumes including peanut, bambara nut, and cowpea, and some tubers. Cropping systems are low-input with limited or no fertilizer application, and farmers typically broadcast seeds. Climate conditions range from a warm desert climate in the West to a warm semi-arid to humid climate in the East (Engelbrecht and Engelbrecht, 2016). Ca. 80% of annual precipitation falls during the cropping season from October to April/May, followed by a May-October dry season. We selected two villages representing the climate extremes for arable farming in the province: Gabaza and Selwana (802 mm MAP and 585 mm MAP, respectively; 2000-2010 period, (Ruane et al., 2015)). Average daily maximum temperatures in January are 30.6 °C and 32.4 °C, and 24.3 °C and 25.9 °C in July for Gabaza and Selwana, respectively (Tab. 1). Both villages are part of research projects (SALLnet, Rötter et al., 2021, https://www.uni-goettingen.de/de/592566.html) and under survey since 2013 (Tab. 1).

**Table 1.** Site characteristics of the two selected study villages and cropland usage (% of cropland area) for the most common crop types according to village surveys conducted in 2019.

| | Unit | Gabaza | Selwana |
|---|---|---|---|
| Coordinates | lat, lon | 23°59'25" S, 30°19'42" E | 23°41'59" S 30°57'03" E |
| Elevation | m a.s.l. | 676 | 372 |
| Average annual rainfall | mm | 802 | 585 |
| Coeff. of variation for annual precip. | | 0.39 | 0.44 |
| Avg. daily mean temp. January | °C | $24.6 \pm 2.2$ | $26.3 \pm 2.3$ |
| Avg. daily mean temp. July | °C | $15.6 \pm 2.3$ | $17.2 \pm 2.3$ |
| Avg. daily max. temp. January | °C | $30.6 \pm 3.3$ | $32.4 \pm 3.6$ |
| Avg. daily max. temp. July | °C | $24.3 \pm 3.5$ | $25.9 \pm 3.5$ |
| Avg. daily min. temp. January | °C | $19.3 \pm 1.9$ | $20.9 \pm 2.0$ |
| Avg. daily min. temp. July | °C | $8.7 \pm 1.9$ | $10.4 \pm 1.9$ |
| Soil texture | | Sandy clay loam | Sandy loam |
| Main ethnic group | | Tsonga | Pedi |
| No. of inhabitants | | 2413 | 5263 |
| Avg. annual household income | ZAR | 54627 | 75585 |
| Formal education aged 20+ | % | 86.7 | 80.1 |
| Water access on homestead | % | 1.5 | 9.6 |
| Distance from paved road | km | <5 | 5-10 |
| Cropland allocation to maize | % | 28.5 | 41 |
| Cropland allocation to cowpea | % | 5.5 | 9 |
| Cropland allocation to bambara | % | 15 | 9 |
| Cropland allocation to peanut | % | 15.5 | 9 |
| Cropland allocation to pumpkin | % | 15 | 8 |
| Livestock units (LU) | | 90 | 87.4 |
| Stocking density | ha/LU | 15.9 | 24.3 |
| Rangeland area size | ha | 1431 | 2128 |
| Rangeland woodland/grassland ratio | | 71/29 | 55/45 |

## 2.2 Current smallholder farming practices versus a minimum sustainable intensification scenario

Farmers in Limpopo have adapted their crop-livestock practices to the strong climatic seasonality. Cattle graze on the communal rangeland during the cropping season. After crop harvest, when rangeland provides little feed, livestock frequently consume the remaining crop residues on-site (Bennett et al., 2009). Sometimes herders destroy fences to give cattle access

to fields, potentially causing conflicts with crop farmers. We characterized village-specific cropping patterns based on a survey and ground-truthing campaign conducted in April and May 2019 and on background information based on working with smallholders in the region for >20 years. Table 1 summarizes the cropping patterns. More than 90% of the farmers individually cultivate <1 ha and typically receive land usage rights from the community (Permission to Occupy, PTO). Maize is the prevailing regional staple (Tab. 1) complemented by legumes, e.g., peanuts and cowpea. Smallholders often recycle maize seeds from the previous harvest. Seeds are broadcasted, planting density is low and frequently ranges between 1-5 plants/m$^2$. While mineral fertilizers are uncommon, cattle feeding on harvest residues accelerates nutrient cycling and concentrates nutrients in dropped manure. Weeds are widespread and sporadically hand-weeded. Farmers do not use agricultural chemicals or machinery. Maize yields are often <1 metric t/ha, with 2-3 t/ha in good years (manure input and regular weeding). We termed this Status Quo of cropland cultivation as the "SQ-scenario".

For the minimum sustainable intensification scenario (minimum SI-scenario), we prescribed 50 kg ha$^{-1}$ yr$^{-1}$ of N-fertilizer application at sowing to increase nutrient supply, but did not consider any phosphorus or potassium applications. Additionally, we implemented weeding during the crop growth period and crop rotation (maize followed by a legume, either peanut or cowpea) to avoid soil exhaustion. APSIM has been tested for this regionally common crop rotation practice (Hoffmann et al., 2020). The frequency of a crop within the rotation was determined by the land allocation of the crop as observed in the field. Cattle had access to the cropland in the dry season (post-harvest) in both SQ and SI scenarios and provided dung input, parameterized as a daily constant input value, that was considered in APSIM simulations as nutrient input source. Dung collection on rangeland and transfer of dung from rangeland to cropland was not included in simulations, as this practice is not common in either study village. This represents a low level of intensification but yet an improvement of the current Status Quo. We deem it realistic that our assumptions for sustainable intensification are feasible for smallholder farmers in the target villages even under current resource constraints, as farmers interviewed in surveys indicated that similar efforts are already made in neighboring villages.

## 2.3 Cattle and rangeland management according to village surveys

During the 2019 survey, local guides approached herders to assess cattle number, age, gender, and breed to establish village-specific feed demand and rangeland usage habits. At Selwana, the rangeland consisted of well-fenced 'herder camps' where herders and cattle lived semi-permanently and animals stay on the rangeland over night. In contrast, herders in Gabaza graze animals during the daytime and return to the homestead at night to reduce the risk of theft. In Gabaza, brief interviews with herders were conducted in the morning before they sent livestock to the rangeland, which also allowed for the visual assessment of the herds. As tracking the herds with GPS collars was not possible, interviewers asked herders to delineate grazing areas on maps. For the parameterization of the rangeland simulations, we determined the daily village-level dry matter demand based on the livestock units (LU), assuming a temporally constant daily dry matter demand of 12.5 kg/LU as aDGVM2 currently cannot quantify temporal changes in biomass quality. For APSIM simulations, we additionally calculated the monthly energy and protein demand of herds based on the surveys to estimate nutrition-based input of manure to cropland. For this, we

parameterized dry matter content, metabolizable energy, crude protein and dry matter digestibility using the values established by Descheemaeker et al. (2018, see their Table 1).

Differing stocking density between villages is represented in aDGVM2 simulations via village-specific daily biomass demand of herds. The differing spatio-temporal distribution of grazing pressure between both villages due to the subdivision of the rangeland area at Gabaza into four spatially separated sub-areas vs. the contiguous rangeland area at Selwana is also considered in aDGVM2 simulations. While we simulated continuous grazing of the rangeland area at Selwana during cattle presence, we explicitly simulated grazing on sub-areas in Gabaza by subdividing the total rangeland grazing time into four periods. However, due to the lack of a sub-daily resolution of the grazing module in aDGVM2, we could not account for differing daytime-nighttime livestock handling between villages.

## 2.4 Coupling APSIM and aDGVM2

We coupled the APSIM crop growth model and the aDGVM2 vegetation model offline. Both models conduct simulations on a daily resolution, and we used the same environmental input data to drive both models. APSIM simulations were conducted prior to the the aDGVM2 simulations to determine (1) crop residue quantities available for livestock during the dry season, and (2) the timing of sowing and harvest to establish the times when cattle could access cropland in the RC scenario. The cropland grazing time windows established through the APSIM simulations were then used to exclude livestock from rangeland in aDGVM2 (no grazing in aDGVM2 simulated during these time windows) in the RC-scenario.

## 2.5 Crop growth simulation using APSIM

We simulated crop growth and development using the Agricultural Production System sIMulator (APSIM v7.9, https://www.apsim.info), which is partly based on early modeling work conducted in Kenya (Keating et al., 2003). Crop growth and development in APSIM are calculated on a daily time step and are affected by temperature, radiation, and water and nutrient supply. APSIM also calculates the dynamics of soil water, nitrogen, and phosphorus on a daily basis. Model outputs include cardinal physiological stages such as duration of flowering and physiological maturity, total above- and belowground biomass, yield, water balance components such as evapotranspiration and soil moisture in different depths of the root zone, nitrogen uptake, nitrate leaching, etc. (Probert et al., 1998; Wang et al., 2014; Whitbread et al., 2017). APSIM was previously calibrated and validated for sites and target crops in the study region, see for example Hoffmann et al. (2018b) and Nelson et al. (2022). These evaluations include a wide range of crops and crop rotations. While Hoffmann et al. (2018b) looked specifically at peanut, Nelson et al. (2022) focused on the calibration of maize for the study sites and two additional villages in the Mopani district, Limpopo. A number of complex crop rotations were tested at sites in South Africa by Hoffmann et al. (2020). For a general overview of APSIM applications in Africa, see Whitbread et al. (2010). For information on APSIM performance regarding different crops and functional aspects, see the studies in Table S1.

## 2.6 APSIM simulation setup

We ran APSIM for the period 1998-10-01 to 2010-10-01 using AgMERRA climate data (Ruane et al., 2015). Sowing took place in a time window between November and December once cumulative rainfall reached 20 mm within a five-day period. Crops were harvested in April/May (two weeks prior to the onset of the animal presence period on cropland in the RC-scenario, see Fig. 1e, f and Fig. 2). We simulated soil water, soil organic carbon (SOC), N, and P continuously between cropping cycles. SOC was halved every 30 cm (see Dagliesh et al., 2016) and simulated to a soil depth of 180 cm, with constant soil texture below the root zone, based on SOC starting values measured during a 2015 field campaign. Simulated crops included maize (sc501), peanut (kangwana), and cowpea (banjo), the most common local types of each crop (see, e.g., Hoffmann et al., 2018b; Rapholo et al., 2019; Hoffmann et al., 2020). We used a generic summer grass and a winter dicot to simulate weeds. Weeds in the SI-scenario were removed at a weed biomass threshold of 2 metric t/ha but at a maximum of 30 days (up to three times during the cropping season) to emulate findings from the ground-truthing campaign (see section 2.2). We simulated two scenarios: i) the average farmers' practice observed in the villages (i.e., "Status Quo", termed SQ-scenario) and ii) a minimum sustainable intensification scenario ("SI-scenario"), according to the specifications defined in section 2.2.

## 2.7 Modeling rangeland dynamics

The aDGVM2 simulates the daily growth and state of individual plants on representative 1-ha stands (Scheiter et al., 2013; Langan et al., 2017). Trait sets describe each individual's growth form (grass, tree, shrub, perennial or annual grass), leaf characteristics (specific leaf area, photosynthetic pathway), carbon allocation to plant compartments, plant architecture (roots, crown shape), response to fire, reproduction, and mortality. Plant performance emerges from trait characteristics and environmental filtering (Scheiter et al., 2013; Langan et al., 2017; Pfeiffer et al., 2019). The model has been specifically developed and tested for conditions in southern Africa (Pfeiffer et al., 2019). To represent grass functional diversity and grazing impacts, the model simulates annual and perennial grasses and accounts for preferential grazing (see model description in Pfeiffer et al., 2019).

### 2.7.1 aDGVM2 simulation setup

The aDGVM2 required animal presence times, the number of livestock units (LUs), and the dry matter demand per LU to determine daily hectare-based biomass removal. We assumed that animals move from cropland to rangeland two weeks before planting and return two weeks after the completion of the crop harvest to feed on crop residues during the dry season. SI or SQ crop management did not influence the timing of rangeland grazing simulations, because the timing of sowing and harvest maximum varied by a few days.

We simulated three different scenarios: 1) Rangeland-only scenario (RO): animals exclusively graze on rangeland; 2) Rangeland-cropland scenario (RC): animals on rangeland during the cropping season, animals on cropland during the dry season; 3) Control scenario (CO): no cattle presence, very low-intensity background grazing on random days (frequency equal to mean annual frequency of RC-scenario, animal density equal to 25% of the mean animal density of the RC-scenario) fol-

lowing the scheme described in Pfeiffer et al. (2019) to ensure the development of a grazing-adapted plant community. Based on test runs, we conducted a 310-year spin-up with a randomized climate data sequence from 1980 to 2010 for all scenarios, followed by a 31-year transient simulation from 1980 to 2010. In the RO- and RC-scenario, we prescribed grazing between 2000 and 2010 and kept the low-level background grazing from the spin-up for the CO-scenario. Natural fire as part of the local rangeland dynamics was allowed during spin-up and transient simulations. Each individual replicate simulation within a scenario was initialized with a unique random seed, creating unique fire occurrence sequences for all simulated hectares. This approach implies small fires that do not fully burn an entire grazing area at a given time and reflects the commonly observed predominance of low-intensity grass layer fires in the region. We used the same set of replicate initializations for all three scenarios, implying that fire event sequences between scenarios were identical up to the start of the grazing treatments, but then deviated because the grazing module also uses random numbers and therefore alters the sequences between different grazing scenarios. As grazing and fire also interact in the field, e.g., via effects of grazing on fuel availability, we consider this variation as an imitation of the naturally occurring effect.

### 2.7.2 Rangeland specifics and animal numbers

Based on the herder surveys (see section 2.3), we set livestock to a total of 90 LU for Gabaza and 87.4 for Selwana. The rangeland area was 1431 ha at Gabaza (15.9 ha/LU) and 2128 ha (24.3 ha/LU) at Selwana. At Gabaza, 71% of the rangeland was woodland and 29% grassland, and at Selwana 55% was woodland and 45% grassland (Tab. 1). While rangeland at Selwana is one contiguous area, Gabaza's rangeland includes four sub-areas (A1 to A4) with area sizes of 57, 279, 394, and 683 ha. Lacking detailed information, we assumed equal woodland-grassland partitioning for all sub-areas.

### 2.7.3 Spatiotemporal distribution of livestock

The aDGVM2 required day-based information of the LU number visiting a given hectare. The aDGVM2 models vegetation on individual 1-ha stands with no information exchange between different hectares. Spatiotemporal sequences mimicking animal movement on grazed areas were created (see following subsections for details) to determine animal effects on each individual hectare on any given day as explicit animal movement was not tracked. At Gabaza, we established an additional temporal subdivision to determine the duration of animal presence on each sub-area. We assumed active herd relocation between sub-areas (and between rangeland and cropland) and prescribed the use of only one sub-area at a time. Additionally, we partitioned presence time proportionally to sub-area size. Therefore, due to this size-proportional time-split between sub-areas, all rangeland hectares experience approximately the same annual demand, independent of their location in a small or large sub-area, but resting periods are longer for hectares located in the smaller sub-areas. Differences in feed deficit size between different sub-areas therefore can be attributed to seasonality, as grazing load is distributed equally between sub-areas. The grazing periods were prescribed as input to the aDGVM2 grazing routine. Figure 2 illustrates presence/absence on rangeland for both villages and scenarios (RO, RC). The proportional reduction of presence time in the RC-scenario compared to the RO-scenario is illustrated in supplementary Fig. S1.

### 2.7.4 Generation of hectare-specific grazing sequences: daily choice of affected hectares

We assigned an index to each hectare in a given area and attributed it to woodland or grassland based on the respective percentages of both vegetation types (Fig. 3a). Then we determined the grazing-affected hectares on a given day by 1) the number of affected hectares ($N_{aff}$) ("how many?") and 2) the indices of affected hectares ("which hectares?"). $N_{aff}$ depends on the herd walking range. Cattle typically move between 1 and 13 km per day (Schrader, 2007). To link walking distance with $N_{aff}$, we assumed that animals moving between hectares walk a minimum of 100 meters (the lateral length of a hectare). Based on the typical range of walking distance combined with the spatial scatter of the herd and lacking more detailed information, we defined a mean of 50 $N_{aff}$ per day, with a standard deviation of $\pm$ 20 ha and a minimum of 10 $N_{aff}$ per day. If the available grazing area was smaller than $N_{aff}$, animals grazed all hectares of the area. We drew a random number from a normal distribution using the 50-ha mean and 20-ha standard deviation to determine daily $N_{aff}$ during animal presence periods on a given area. If a random number was < 10, we replaced it with a random-uniform number between 10 and 100 hectares to ensure the minimum of 10 $N_{aff}$ per day. In this way, we created a frequency distribution for the number of visited hectares per day akin to the one shown in Fig. 3b). Knowing $N_{aff}$ for each day in the 11-year simulation period, we determined the daily indices of affected hectares ($I_{aff}$) via daily random-uniform subsampling from all available hectare indices of the considered area (Fig. 3c), i.e., by randomly choosing $N_{aff}$ random indices.

### 2.7.5 Daily animal distribution across affected hectares

To distribute the LU across affected hectares $I_{aff}$, we converted LU to daily animal units (AU) by multiplying the minutes per day with the number of LU (i.e., 1440 x 90 LU = 129000 AU for Gabaza, and 1440 x 87.4 LU = 125856 AU for Selwana). The minute-resolution considers that animals may spend only part of a day on an individual hectare. For each day, we randomly created between 1 and 5 animal subgroups ($0<N_g<6$) with varying sizes and spatial densities (Fig. 3d). We placed the group centers randomly on the $N_{aff}$, as shown in Fig. 3e for an example with 5 sub-groups. For each group, we drew the standard deviation around the group center from a random uniform distribution ranging between 0.5 and 1 times $N_{aff}$ (colored horizontal lines in Fig. 3e illustrate the standard deviation, the vertical lines the locations of the individual group centers). We chose this spread around the group center somewhat arbitrarily to describe how tightly the animals within a group stay together. The scatter also influences how strongly subgroups intermingle on the number of affected hectares. We assigned the AU to the different groups by creating fractional group sizes by first drawing ($N_g$-1) random uniform numbers between 0 and 1, then size-sorting these to establish the group breaks. Fractional group sizes were calculated as the differences between the breaks (including 0 and 1 as lower and upper edges) and multiplied with the number of AU to obtain the daily AU per group (Fig. 3f). For each AU group, we applied a random normal distribution using the group's mean and standard deviation (group center and scatter, as shown in Fig. 3e) to determine how many AU are assigned to the hectares around the group mean. We iteratively re-distributed AU whose drawn numbers were <1 or >$N_{aff}$.

Generating individual distributions for varying numbers of animal groups of changing size allows flexible animal distribution across the affected hectares. The super-ordinate animal distribution of the herd emerges from superimposing the individual normal distributions (Fig. 3g) and allows the creation of flexible multi-modal livestock distributions. It imitates the daily livestock group dynamics in a pseudo-explicit manner that does not require exact knowledge of spatial relationships between affected hectares while still creating average long-term characteristics of herd behavior and rangeland usage. When assigning the AU/ha to the affected hectares, we sorted the AU/ha in descending order and randomly assigned the lower range of the ordered list to the affected woodland hectares to account for the higher feed availability on grassland than woodland. For use in aDGVM2, we reconverted the daily assigned AU for each affected hectare to daily LU per hectare (Fig. 3g).

### 2.7.6 Simulation of representative hectares

Although we created grazing sequences for all hectares (see supplementary material), computational constraints precluded simulating the total rangeland. We therefore simulated different grazing intensity levels based on total LU visits/ha during the experiment. For each rangeland (sub-)area, we conducted 150 simulations (75 grassland, 75 woodland hectares), with 15 simulations per area and vegetation type ranging around the minimum, maximum, median, 25-percentile and 75-percentile of cumulative LU-numbers/ha over 11 years, respectively. For sub-area A1 at Gabaza, all 57 ha, i.e., the complete sub-area, was simulated. For simulated percentages of (sub-)areas, see Fig. 1a. We conducted 657 simulations for each scenario (CO, RO, RC), i.e., 1971 simulations total.

### 2.7.7 Feed gap analysis

Consumable grass biomass includes living leaves and stalks, standing dead and reproductive biomass. A minimum amount of each pool is unavailable to grazers as they cannot graze completely down to the ground. For cattle, we defined a limit of 30 $g/m^2$ of living and dead biomass that needs to remain, respectively, and 10 $g/m^2$ of seed biomass (see Pfeiffer et al., 2019). Feed gaps occur when demand exceeds available biomass. We calculated average annual demand, consumption, and deficit across all simulated hectares, by vegetation type (i.e., for woodland and grassland hectares), for both villages, and (sub-)areas. In addition, we summed the annual deficits of all simulated hectares per village and divided this sum by the summed demand from all simulated hectares. This ratio characterizes demand-specific deficit severity and allows comparisons between different years. In the same way, we calculated the ratio between the number of days with a grazing deficit and the sum of rangeland grazing days. As animals in the RC-scenario only spent part of the year on the rangeland, we scaled both ratios by the fraction of the year animals were present on the rangeland for this scenario. We also created hectare-scale overviews for biomass consumption and demand to assess variability between hectares, which allowed for comparisons between both villages, as well as sub-areas at Gabaza.

## 2.8 Performance indicators

For the evaluation of the cropland simulations, we analyzed the temporal grain and straw yield (dry matter) dynamics at both villages and determined site-specific differences in yield patterns and between SQ- and SI-scenarios. Aside from yields, we analyzed SOC status, cropland vegetation cover, and soil-water dynamics. For the rangeland simulations, we considered metrics that describe animal-related aspects and grazing pressure on vegetation, such as grazing frequency and biomass demand relative to consumption. We identified when feed gaps occur and their severity compared to demand. Additionally, we examined the seasonal and interannual dynamics of grass biomass and productivity (NPP, GPP) to determine between-village difference between the RO- and RC-scenarios.

## 3 Results

### 3.1 Cropland simulation results

#### 3.1.1 Yield comparison between the two villages

**Maize grain yield**

Village-scale grain yields showed considerable interannual variability (Fig. 4a and b; see Fig. S2 for individual depiction per crop type). With a larger arable area, Selwana had higher total yields than Gabaza but lower yields per hectare due to the drier conditions (Fig. 4c and d). The total yield at Gabaza in the SQ-scenario varied between 396 metric t in 2005 and 1253 t in 2006 (average: $910\pm251$ t). Yields per hectare ranged between 1.2 metric t/ha and 3.7 t/ha (average of $2.7\pm0.8$ t/ha). SI led to a moderate, non-significant average yield increase by factor $1.2\pm0.1$ (Fig. 4f). Straw yield was moderately, but at Selwana not significantly, increased by SI (Fig. 4e). Maize village production at Selwana varied greatly, from 367 metric t in 2001 to 5739 t in 2004 (average: $2620\pm1603$ t). Maize grain yields per hectare at Selwana were $53\pm23\%$ lower than at Gabaza, ranging between 0.2 t/ha and 2.9 t/ha (average of $1.3\pm0.8$ t/ha). Compared to Gabaza, SI at Selwana led to a slightly higher average maize yield increase (by factor $1.2\pm0.3$, statistically non-significant), but the response varied considerably between years (Fig. 4f). The highest yield increase at Selwana (factor 1.8) occurred in 2005. However, SI reduced maize yield in 2004 by 11% compared to the SQ-scenario, and to a lesser extent also in 2002 and 2006. At Gabaza, SI consistently increased maize yields.

**Cowpea grain yield**

The mean cowpea yield in the SQ-scenario at Gabaza was $62\pm24$ t, with a minimum of 30 t in 2004 and a maximum of 96 t in 2007, and a mean yield per hectare of $0.7\pm0.4$ t/ha (Fig. 4). In the SI-scenario, grain yield on average increased by factor $1.6\pm0.0$ (statistically significant increase at $p<0.05$; Fig. 4f). The total mean village cowpea grain yield in the SQ-scenario at Selwana was $169\pm92$ t, with a minimum of 36 t in 2005 and a maximum of 328 t in 2008. By hectare the average yield was $0.4\pm0.2$ t/ha (minimum: 0.1 t/ha; maximum 0.7 t/ha). SI led to an average non-significant yield increase by a factor of $1.3\pm0.2$.

**Peanut grain yield**

Gabaza had a village-scale peanut yield of 161±27 t, with a minimum of 125 t in 2005 and a maximum of 214 t in 2000
(Fig. 4b). Average yield per hectare was 0.9±0.2 t/ha (minimum: 0.7 t/ha; maximum: 1.2 t/ha; Fig. 4d). SI increased the yield
by an average factor of 1.2±0.0 (range: 1.1 to 1.3, increase significant at p<0.05; Fig. 4f). At Selwana, the village-scale peanut
yield was 199±74 t, ranging between 63 t in 2003 and 309 t in 2000 (Fig. 4a). By hectare, the average yield was 0.5±0.2 t/ha
(minimum: 0.1 t/ha; maximum 0.7 t/ha; (Fig. 4c). On average, SI increased peanut yields at Selwana by a factor of 1.3±0.3
(statistically non-significant), with the strongest positive effect in 2001 (factor 1.7 increase; Fig. 4f). However, in 2005 yield at
Selwana was reduced by more than 50% compared to the SQ-scenario.

**Effect of SI measures on grain yields - summary comparison**

For cowpea, SI had a stronger positive effect at Gabaza for relative and hectare-specific increases. Cowpea benefited the
most with a factorial increase of 1.6±0.0 (absolute yield increase per hectare of 0.6±0.2 t/ha). With the highest hectare-
specific yields of all crops, the moderate increase by a factor of 1.2±0.1 led to an overall increase of 0.4 t/ha for maize, while
the relative increase by a factor of 1.2±0.0 for peanut corresponded to a yield increase of 0.2±0.1 t/ha. At Selwana, hectare-
specific yield increase was highest for maize (0.2±0.4 t/ha; relative increase by factor 1.2±0.3), followed by peanut (1.2±0.1
t/ha; relative increase by a factor of 1.3±0.3) and cowpea (0.1±0.1 t/ha; relative increase by a factor of 1.3±0.2). The crop-type
specific responses also caused a slight shift in the relative contribution of each crop type to the total village-level yield (see Fig.
S3).

SI led to a statistically significant increase in village-scale yield at Gabaza for cowpea and peanut, but not for maize (Welch
two-sample t-test, p<0.05). No crop type at Selwana showed a significant increase. Moreover, interannual variability of SI-
related yield effects was more pronounced and even negative in some years.

### 3.1.2 Soil organic carbon (SOC), cropland vegetation cover, and soil water

Cropland SOC gradually decreased irrespective of management (Fig. 5), but somewhat less under SI. At Gabaza, SOC declined
by 4.7% from 7.5 kg C/m$^2$ to 7.2 kg C/m$^2$ in the SQ-scenario, and by 3.7% in the SI-scenario (7.3 kg C/m$^2$ left in 2010). On
average, cropland soil at Selwana stored $1.4 \pm 0.1$ kg C/m$^2$ less than at Gabaza, and C-loss was 6.2% in the SQ-scenario (from
6.1 kg C/m$^2$ in 1998 to 5.7 kg C/m$^2$ in 2010). SI only had a minor effect on SOC-loss at Selwana (5.6% loss, 5.8 kg C/m$^2$
in 2010). At the village scale, cropland soils under SI had 427 (Gabaza) and 1030 (Selwana) tons more carbon stored in 2010
than under SQ.

Average green leaf area index (LAI) on cropland varied seasonally and interannually (Fig. 6). Monthly values at Selwana
(Fig. 6a) were generally lower than at Gabaza (Fig. 6b). SI had a moderately positive but statistically non-significant effect on
LAI, and the highest canopy-cover values occurred during the growing period from December to March. After crop harvest,
weed infestation drove cropland LAI during the dry season.

Extractable soil water (ESW) was generally higher at Gabaza (Fig. 6c, d). Minimum values occurred in August and Septem-
ber and increased throughout the wet season before gradually declining again. At both locations, SI tended to reduce ESW due

to the higher vegetation cover and biomass simulated in the scenario. However, the difference between the SQ- and SI-scenario was non-significant.

## 3.2 Rangeland simulation results

### 3.2.1 Biomass demand, grazing frequency, and consumption

Hectare-specific average annual grass biomass demand was lower at Selwana (Figs. 7, 8; for detailed numbers, see Tab. S2a). There, animals on average required approx. 300 kg/ha/yr from grassland hectares in the RO-scenario, but only between 70 and and 190 kg/ha/yr in the RC-scenario. The biomass demand on woodland hectares was lower in both scenarios due to animals preferentially visiting grassland hectares with higher biomass availability. Due to the higher stocking density, the annual hectare-specific demand at Gabaza was consistently higher than at Selwana (see panels b-e in Figs. 7 and 8), roughly ranging between 470 and 630 kg/ha/yr in the RO-scenario and between130 to 340 kg/ha/yr in the RC-scenario on grassland hectares. Due to the presence period scaling being proportional to the size of the sub-areas at Gabaza, the hectare-specific demand is approx. equal for all sub-areas. Interannual variability in average demand results from the variability in the visit pattern generated according to the description in section 2.7 and the fact that only a sample of the total number of rangeland hectares was simulated at each site.

On average, grazers visited each hectare at Selwana on 8-9 d/yr in the RO-scenario (Tab. S2b), and on to 2-6 d/yr in the RC-scenario. At Gabaza, LU on average frequented each hectare on 11-14 d/yr in the RO-scenario and on 3-8 d/yr in the RC-scenario, except for the smallest sub-area A1 where the average visit frequency was between 7-11 d/yr in the RO-scenario (annual presence on A1: 15 d/yr), and between 3-6 d/yr in the RC-scenario (presence on A1: 3-8 d/yr, depending on the year).

Despite a similar visit frequency, the average grass biomass consumption on woodland was lower than on grassland (Tab. S2c) due to the lower number of animals assigned to woodland hectares (see section 2.7.5). Average daily grass biomass consumption on affected hectares was comparable between the RO-scenario and RC-scenario as on a per-day-basis the same number of animals per area was present during both scenarios. Differences in annual grass biomass consumption between scenarios only emerged due to the shortened presence times per area in the RC-scenario (Tab. S2d).

### 3.2.2 Feed gaps on rangeland

Across all simulated hectares, the annual demand-specific deficit (gap between supply and demand) at Selwana ranged between 0.2 and 11.4% in the RO-scenario (Fig. 9a). In the RC-scenario, the deficit declined to values between 0-5.1%, partially due to the shorter herd presence on the rangeland. Gabaza, where the average cattle density per ha was 1.5 times higher, had higher relative deficits. Moreover, Gabaza has a higher relative woodland proportion (Tab. 1). Demand-specific deficits at Gabaza were between 0.4 and 17.5% in the RO-scenario (Fig. 9b) and declined to values between 0-8.0% in the RC-scenario. Years with higher deficits, such as 2002, 2003, 2005, and 2008, coincide with low annual precipitation at both sites (Fig. 1b). The maximum deficits occurred in 2003, the second dry year in a row after 2002. For relative deficits separate for grassland and woodland hectares, see Fig. S4.

Integrated across all simulated hectares, the timeline of the percentage of grazing days with a deficit resembles the one for relative deficits (compare Fig. 9c and a, and d and b). Between 0.6 and 11.5% of grazing days had a deficit in the RO-scenario at Selwana. The RC-scenario reduced this to a range from 0.0 to 5.2% of total days. At Gabaza, the higher grazing pressure caused more deficit days. Here, the RO-scenario resulted in a range from 0.5 to 17.0% of grazing days, with a reduction to 0.0-6.6% in the RC-scenario. For a presentation by vegetation type, see Fig. S5. Deficit timing showed a clear seasonal pattern (Fig. 9e, f). Monthly deficits in the RO-scenario were lowest between March and June and highest in October/November before declining towards December. While both villages had a similar seasonal pattern, the demand-specific monthly deficits were higher at Gabaza, particularly from September to November. Cropland pasturing between ca. April to October (see Fig. 1e, f) avoided deficits during these months. However, it also reduced the deficit size during rangeland presence (compare scenario RO vs. RC in Fig. 9e, f). Deficits showed a delay of 2-3 months relative to precipitation (Fig. 1c, d), i.e., the maximum occurred delayed after the end of the dry season, when precipitation increased again. Similarly, the lowest deficits in March /April occurred ca. 2 months after peak precipitation.

### 3.2.3 Temporal dynamics of grass biomass

Consumable grass biomass provision was distinctively seasonal (Figs. 10, 11). Peak biomass occurred between February to March, and minimum availability developed towards the end of the dry season in October/November. Overall biomass availability was lower at Selwana than at Gabaza, and woodland hectares produced less consumable biomass than grassland hectares. Consumable biomass also greatly varied between years. As the second of two consecutive dry years, 2003 had the lowest biomass from the 11 years. Grazing had a minor effect on across-hectare average consumable biomass (compare scenario averages and standard deviations in Figs. 10, 11). Differences in fire occurrence between grazing scenarios also caused biomass differences (see Figs. S6 and S7).

### 3.2.4 Rangeland productivity (NPP, GPP)

For Selwana grassland, the average annual GPP in the control ranged between 3.1±1.5 to 11.2±4.1 t/ha, whereas Gabaza had values between 3.7±1.9 to 15.8±6.5 t/ha (Tab. S3, Fig. S8). On woodland, hectare-specific grass-layer GPP was lower and ranged between 1.2±0.6 to 3.6±1.7 t/ha at Selwana, and between 1.3±0.7 to 4.7±1.6 t/ha at Gabaza (Tab. S3, Fig. S9). Differences between scenarios were statistically non-significant (two-sided t-test, $p<0.05$).

Annual grassland NPP at Selwana varied considerably and ranged between 1.2±0.5 to 6.0±2.3 t/ha in the control scenario. Annual grassland NPP at Gabaza was higher, with values between 1.4±0.6 to 8.7±3.8 t/ha (Tab. S4, Fig. S10). We also simulated this pattern for woodland hectares, where average annual NPP ranged between 0.4±0.3 to 1.9±0.9 t/ha at Selwana, and between 0.5±0.3 to 3.0±0.9 t/ha at Gabaza (Tab. S4, Fig. S11). The RO- and RC-scenario did not differ significantly (two-sided t-test, $p<0.05$) from the control, except for the years 2005 and 2007 in the RC-scenario for Selwana grassland hectares, and the year 2008 in the RO-scenario for the woodland hectares of sub-area A1 at Gabaza.

GPP normalized for living leaf biomass (LLBM) had annual values approximately between 7 and 13 g C/g LLBM (see Tab. S5 and Figs. S12, S13), and between 2 and 8 g C/g LLBM for normalized NPP (see Tab. S6 and Figs. S14, S15). Values of

biomass-specific GPP and NPP were comparable between grassland and woodland, i.e., grass on woodland was as productive as grass on grassland. Different from the hectare-specific GPP and NPP values, grazing frequently caused significantly (two-sided t-test with $p<0.05$) higher average biomass-normalized GPP and NPP values relative to control (see Figs. S12, S13, S14, and S15). This effect was usually stronger in the RO-scenario. Based on the sensitivity study in Pfeiffer et al. (2019), we suspect that an optimum grazing frequency and intensity likely exists where simulated biomass-normalized productivity is maximized, but we did not explicitly test for this optimum in either the cited or present study. Plot-level GPP and NPP showed pronounced seasonality, and monthly values varied strongly between years (Figs. S16, S17, S18, and S19).

We saw similar seasonality for biomass-specific monthly GPP and NPP (Figs: S20, S21, S22, S23) with minimum values in June and a gradual increase to the December maximum. Values gradually declined from January to March and rapidly towards June. Integrated across all simulated hectares, annual biomass consumption relative to NPP varied between years and was lower at Selwana (Fig. 12). The highest ratios occurred in the dry years 2002, 2005, and 2008 (Fig. S10, S11). The annual consumption/NPP ratio was approximately halved in the RC-scenario compared to the RO-scenario (reduction by a factor of $2.1\pm0.6$ at Selwana; $2.1\pm0.2$ at Gabaza).

## 4    Discussion

The UN Sustainable Development Goals (SDGs; UN General Assembly, 2015) include alleviating poverty (SDG1), reducing hunger and enhancing food production (SDG2), and reducing habitat loss and degradation to preserve terrestrial ecosystems and biodiversity (SDG13). These are among the most desirable goals for rural areas in southern Africa. Agricultural scientists emphasize SI as a way to improve cropland productivity (e.g., Mueller et al., 2012; Cassman and Grassini, 2020) while reducing environmental impacts and maintaining ESS and ESF (Tilman et al., 2011; Tscharntke et al., 2012). Given that smallholder farming provides more than 80% of the food supply in sub-Saharan Africa and Asia (Walpole et al., 2013), SI measure adaptation in smallholder farming systems needs a special focus and an integrated system approach with a range of possible management interventions (Vanlauwe and Dobermann, 2020). With more than 500 million smallholder farms worldwide sustaining the livelihoods of more than two billion people (Walpole et al., 2013), smallholder farming needs explicit consideration when discussing SI measures. Landscape-based integrative adaptations linking agricultural and natural systems are required to ensure the continued provision of ESS and help smallholders adapt to climate change (Harvey et al., 2013; Vignola et al., 2015). In our study, we present an example that illustrates how linked crop and rangeland modeling can address research questions on the sustainable management of smallholder farming systems at the landscape level. We focused on research questions revolving around (i) the effect of minimum levels of intensification on crop production, (ii) the trade-offs and opportunities of combined versus separated management of cropland and rangeland for cattle production.

### 4.1    Impact of minimum SI measures on cropland ESS

The minimum intensification measures implemented in our simulations had minor to moderate effects on cropland ESS. Carbon sequestration increased moderately due to higher yields and SOC input. While SI-measures increased SOC sequestration

compared to the SQ-scenario at the end of the 11-year simulation period, SI could not reverse the decreasing trend and cropland soils remained a carbon source, indicating yet too low input of organic material to the soil. Due to the rough parameterization of soil SOC in APSIM, where C-conents are halved with every additional 30 cm of soil depth, SOC losses in kg/m$^2$ over time may be approximate, but should nonetheless capture the relative trends and differences between scenarios. Although SI had a negative effect on the amount of plant-available soil water, the interannual variability was larger than the difference caused by SI.

We expected a stronger relative improvement through SI at Selwana, where SQ produced lower yields per hectare compared to Gabaza, and therefore the potential for improvement seemed particularly promising. However, a general prediction of SI effect strength turned out as challenging due to the variety of influencing factors that may enforce, but in most cases counter, SI effects. In particular, more pronounced water availability constraints at Selwana under SI caused great interannual response variability. Moreover, crop type-specific response intensity was not consistent between sites. With an average significant increase of 59%, cowpea yields at Gabaza reacted most strongly to SI. Weeding, fertilizer and manure input moderately improved productivity when soil water was not limiting, with potential for higher increases with yet more nitrogen and additional phosphorus input (the latter often a major limitation factor in the province's smallholder fields). While N-limitation may be less critical for N-fixing legumes such as peanut and cowpea than for maize, both crops nonetheless responded positively to N-input, showing that the reduced energy expense required for N-fixation also improved legume growth. However, at Selwana, SI measures could result in reduced yields in some years compared to SQ due to enhanced crop growth causing more severe water deficits during crucial times of the growing season. Improved nutrient provision underSI boosts early-stage crop development and results in increased biomass accumulation compared to SQ without fertilizer input. During later stages, the augmented biomass requires more soil water, which can reduce yields and even cause crop failure in years where water becomes limited during later crop development stages. Without irrigation, crops cannot benefit from the additional nutrients from soil amendments under water limitation. Targeted deficit irrigation could make SI measures more effective but is often unavailable to smallholders. From our survey work, two out of 140 smallholders irrigated. However, some initiatives show promise in making irrigation feasible for smallholder farmers, for example through public investments in South Africa, public-private partnerships in Zimbabwe, and SI business models in Tanzania (Hanjra and Williams, 2020). Whether irrigation is feasible also depends on regional water availability. Recent investigations in our project area (e.g., Lam et al., in review) have shown a decline in available surface and groundwater resources in some of the catchments. However, very restricted deficit irritation is still often possible if water is collected from rainwater harvesting, and if irrigation is realized as drip irrigation from available boreholes and surface water without exhausting water resources, as reported in Magomeyi et al. (2018) and Parry et al. (2020). According to Parry et al. (2020), a combination of training and experiential learning of farmers in the context of irrigation can lead to significant change, including water use reduction, improved nutrient retention, and greater yields.

Plant water availability also depends on soil water holding capacity (SWC), which is determined by soil texture, SOC, soil flora and fauna, and soil structure. Particularly on sandy soils, organic matter substantially improves SWC. In addition, loosening compacted soils is an effective way to increase infiltration, create a favorable structure, and increase SWC. The sandy clay loam at Gabaza and the sandy loam at Selwana may specifically benefit from loosening where the clay and silt components

make the substrate prone to slake, hard-setting, and surface crusting, which can cause poor water and air filtration and increase erosion risk. While farmers have little influence on soil texture, they can nonetheless improve soil structure with considerable SWC-enhancing effects (Suzuki et al., 2007). Another measure to amend soil water infiltration and storage is to increase soil organic matter (SOM) by adding plant or animal material, which in turn also reduces soil erosion (Mohler and Johnson, 2009). In addition, utilizing certain aspects of conservation agriculture (CA) practices, such as diversified crop rotations and crop residue retention can also enhance SOM development, crop yields, and climate resilience (Franzluebbers, 2002; Lehman et al., 2017; Williams et al., 2018; Hoffmann et al., 2020). Participatory research with farmers could help identify realistic pathways for SOM-enhancing interventions to be included in future simulations that explore SI for smallholder systems.

## 4.2   Feed gaps at village-level and village-specific differences

Feed gaps occurred in both villages. At Gabaza, higher MAP yielded higher productivity and biomass availability for grazers than the arid rangeland at Selwana. However, higher cattle density at Gabaza compared to Selwana implied higher grazing pressure, which counteracted more favorable environmental conditions and caused higher average deficits at Gabaza.

On average, a given rangeland hectare was grazed on very few days a year (Tab. S2b). This implies that there was ample time for biomass recovery. Additionally, average hectare-specific consumption was low compared to productivity and had no severe impact on biomass, GPP, and NPP (Figs. 10, 11, S8, S9, S10, S11). These indicators do not hint at a general overgrazing problem, where we would expect a drastic reduction of biomass and rangeland productivity. Moderate rangeland grazing stimulated biomass-specific grass NPP by triggering biomass re-growth and growth overcompensation. Grazing is also beneficial because it removes dead biomass, reduces the LAI of living grasses and self-shading within the grass layer, and increases leaf area-specific productivity. Re-growth can also indirectly increase leaf-specific productivity because young leaf tissue tends to be high in nitrogen and photosynthetically more active than old leaves (Kitajima et al., 1997; Mediavilla and A., 2003). However, aDGVM2 does not currently capture this last effect. The effect of grazing intensity and frequency on pasture regeneration was also not explicitly investigated in this study, but is subject of a separate study on the effects of drought and grazing that is currently in preparation (Behn et al., in prep.).

Due to the lack of explicit spatial movement patterns and information on preferred cattle resting and grazing places, we likely underestimated imbalances in rangeland usage, i.e., we may not have captured overgrazing on specific areas, e.g., next to resting or watering places. Here a long-term monitoring of cattle movement using GPS collars (Bailey et al., 2018) could reveal such patterns and allow for their incorporation in rangeland simulations.

Substantial feed gaps despite little biomass and productivity reductions seem contradictory. Feed shortage timing explains how deficits occur despite moderate stocking densities. Agreeing with herder statements, we simulated the largest shortfalls at the dry-to-wet season transition (Fig. 9e,f). This also agrees with the findings of Lamega et al. (2021) that mixed crop-livestock farmers in the drier parts of Limpopo perceive spring as the time with the most pronounced feed gaps. In most years, biomass shortage started in August and intensified until October/November before gradually declining. The mid to late wet season and early dry season usually experienced minimal deficits. Therefore, feed shortages occurred with 2-3 month delays relative to precipitation seasonality, i.e., they were most prominent when precipitation started increasing, and lowest in March/April

approximately two months after peak precipitation. This pattern likely emerges because vegetation growth starts shortly after the onset of the first rains, but peak biomass is reached with a delay of 2-3 months after the start of the wet season. During the time when biomass in-growth happens, grass biomass quantities are a) not yet sufficient to fully supply the demand, b) dead biomass has been largely consumed during the preceding dry season, and c) grazing additionally slows the development of new grass biomass. Therefore, livestock are most prone to experience deficits during this critical time.

## 4.3 Closing feed gaps with integrated management of cropland and rangeland

Our results show that dry-season residue grazing on cropland can reduce feed shortages. Dry-season access to cropland often more than halved the annual feed gap and shortened the feed scarcity period. Moreover, dead grass on the rangeland lasted longer into the early wet season, which reduced the feed gap between September and November. Eliminating feed gaps would require cattle to return to the rangeland at a later stage than simulated, i.e., when early re-growth of grasses has terminated. However, a later return would collide with crop planting dates. For sufficient feed supply during the transition period, we therefore recommend storing part of the crop residues directly after harvest, given availability of storage space and appropriate storage possibilities. This practice is common in other densely populated areas of Africa from Senegal to Ethiopia. The stored residues can then feed cattle until rangeland provides sufficient biomass. Controlled feeding also allows for the treatment of stover to improve nutrient and crude protein supply, to increase feed intake and reduce dry-season weight loss in cattle (Smith, 2002). Digestibility, energy value, nutrients, and protein content also vary with crop residue type. Therefore, residue mixing can additionally improve livestock supply. Ideally, livestock farmers should provide licks and ensure gradual adaptation to residues to avoid typical problems associated with crop residue feeding (Hofmeyr, 2020). In this context, well-trained extension staff who provides such information for farmers is crucial.

## 4.4 Identification of management strategies for sustained provision of ESF and ESS at the landscape-level

Linking cropland and rangeland modeling helped analyze individual management scenarios for both land use types and allowed for the identification of management strategies that maximize benefits at the landscape level. In this study, a comparison of village level crop residue quantities with the feed gap from rangeland grazing revealed how much of the dry season feed gap can be closed with residue grazing. However, consideration of time-dependent aspects also showed that quantities are only a partial aspect that may lead to incomplete conclusions. The timing of the most severe feed deficit at the beginning of the wet season coincides with the start of the crop planting season. Therefore, a complete elimination of feed deficits requires off-field access to previously collected crop residue. Moreover, feed quality related aspects may be underestimated in our study, as aDGVM2 only crudely accounts for quality by assigning dead biomass a nutrition value of 66% compared to living grass biomass, but does not track seasonal changes in living grass biomass.

Dry-season crop residue grazing can reduce feed gaps and accelerate nutrient turnover via manure dropping. The simulated village-level crop residues provided feed supply and additionally allowed SOM formation on cropland. However, neither the SQ- nor the SI-scenario had enough carbon input to reverse the decreasing SOC trend on cropland, but SI-measures reduced SOC loss rates. Although we did not conduct crop growth simulations without cattle residue grazing, the comparably low yields

and nutrient input levels suggest that avoiding SOC-loss may be challenging even without cattle presence. Implementation of context-specific conservation agriculture measures may improve cropland SOC formation and soil water availability.

Based on our results, we propose the following strategies to ensure the sustained provision of landscape-level ESF and ESS: (1) Apply the minimal SI measures prescribed in our simulations and considered feasible for smallholders at our study villages to moderately increase yields of the staple crops. To avoid the negative effects of SI measures due to water limitation, the adoption of deficit irrigation would be ideal, combined with water conservation measures, runoff prevention, rainwater harvesting, and soil amendments to increase soil water storage capacity, in particular where irrigation is not possible. (2) To reduce dry-season grazing deficits, we propose to give cattle access to cropland and allow crop residue grazing. At the beginning of the crop planting season, post-harvest collected crop residues can be offered off-field to avoid animal deficits and allow for a quick build-up of new grass biomass by alleviating grazing pressure on the rangeland.

## 5 Conclusions

Integrated crop-livestock management of highly diverse smallholder farming systems in semi-arid ecosystems is crucial to ensure the continued provision of ESF and ESS. Here, the presented linked cropland-rangeland simulations can identify potential pathways towards the SI of crop production and reduction of livestock feed gaps. We found that modest SI measures – deemed feasible for the smallholder farmers in question - can increase yields and SOC sequestration; yet, water limitation during later-stage crop development can counteract SI measures without adequate irrigation or measures to conserve water and increase soil water holding capacity. We found that dry-season cropland residue grazing can substantially reduce feed deficits. However, the most severe rangeland feed gaps occurred at the beginning of the wet season when grass biomass re-growth was at an early stage. Both findings have marked implications at the policy level and call for appropriate actions. If crop residues are abundant, partial post-harvest collection, storage, and provision of residues as feed during the dry-to-wet season transition period could reduce grazing pressure during early rangeland vegetation development. Research approaches that capture landscape-level interactions and synergies are crucial as climate change impacts and extreme events become more severe. The future climate resilience of ESF and ESS needs a landscape-scale evaluation to identify effective mitigation and adaptation strategies. Targeted experimental work and environmental monitoring allows for the evaluation of model components for new scenarios. Subsequently, an updated integrative modeling framework as suggested by Rötter et al. (2021) can be applied to test different climate change and management scenarios. Further development should include an animal physiology model to simulate the dynamics of animal growth, reproduction and health condition based on nutrition status. Furthermore, incorporating agent-based modeling could improve the representation of animal behavior on the rangeland and account for interactive decision-processes made by human agents based on both economic and ecological criteria. Likewise, further management options could be included, such as the identification and sowing of suitable "dry season cover crops" that are beneficial for animal nutrition and soil improvement.

*Competing interests.* The authors declare that they have no conflict of interest.

*Author contributions.* MP, MH, RR and SS conceived the study. MP and MH conducted the aDGVM2 and APSIM simulations. MP, MH and WN conducted the analysis of the simulation results. MP led the writing of this article. MH, WN, JI, KA, JO and RR were involved in
and contributed to the field survey campaigns and data collection. All authors contributed to the writing of this article.

*Code availability.* The presented data and data analysis scripts used to conduct the data analysis and to create the figures shown in the paper and the supplementary material are available at https://doi.org/10.17605/OSF.IO/R2TS7. The aDGVM2 code contains currently unpublished parts not related to this publication and is available upon request from the authors.

*Financial support.* This work was conducted within the South African Limpopo Landscapes Network - SPACES2: SALLnet project (grant
numbers: 01LL1802A, 01LL1802B) funded by the German Federal Ministry of Education and Research (http://www.bmbf.de/en/).

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

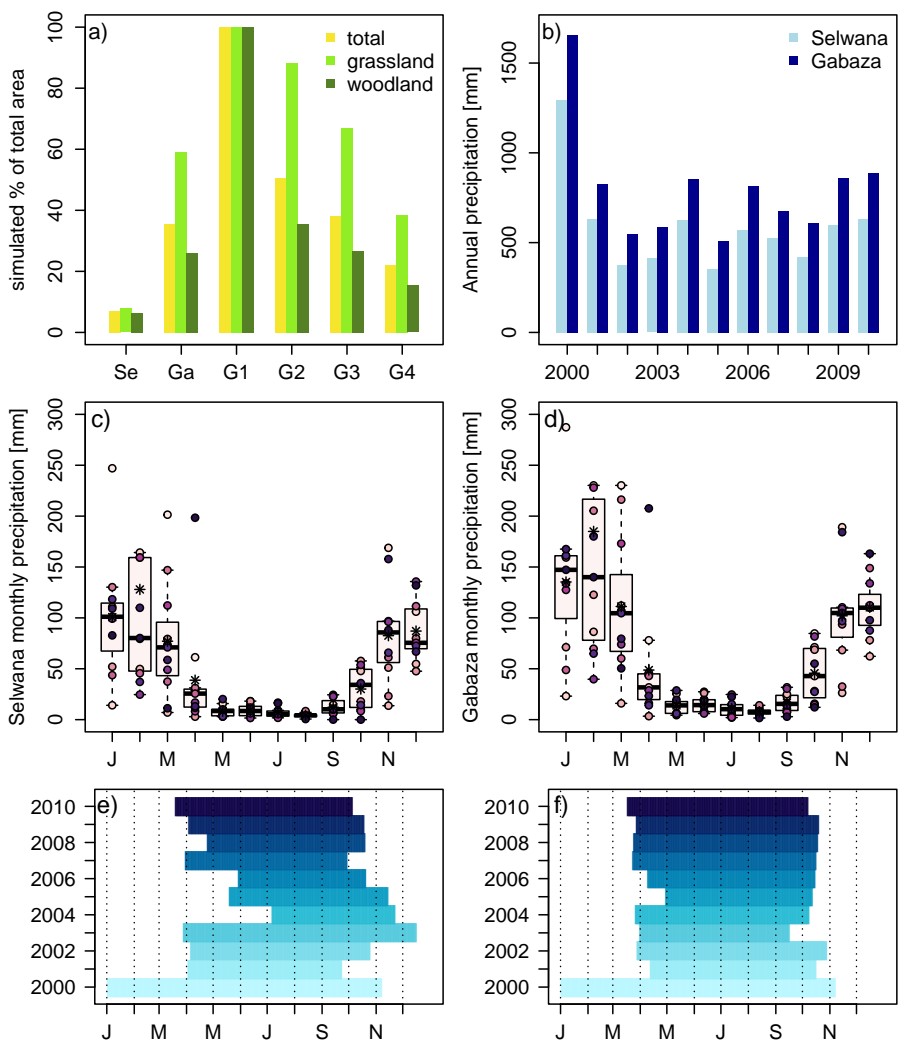

**Figure 1.** Simulated area percentages for the different sites and (sub-)areas (panel a), and annual precipitation for the years 2000-2010 for Selwana and Gabaza from the AgMERRA climatology used to drive APSIM and aDGVM2 simulations (panel b). Panels c) and d) show monthly precipitation at Selwana and Gabaza, respectively, where stars indicate the 2000-2010 average, the dots the individual annual values. Panels e) and f) show the annual timelines of animal presence on cropland in the RC-scenario. A maximum of 150 hectares (75 grassland and 75 woodland hectares) was simulated per (sub-)area, amounting to the shown simulated percentages of total area. Abbreviations: SE: Selwana; Ga: Gabaza; Numbers for Ga indicate sub-areas. For sub-area Ga1 (total size 57 ha) we simulated all individual hectares of the sub-area.

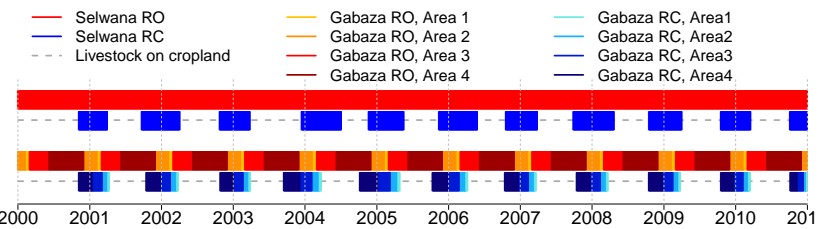

**Figure 2.** Livestock presence times on communal rangeland. Red-hued bars illustrate times of rangeland presence in the RO-scenario, blue-hued bars show rangeland presence/crop growth period in the RC/SQ-scenario, dashed grey lines indicate animal presence on cropland. For Gabaza (bottom pair of bar sequences), the different color hues show how animal presence on the four sub-areas. Animal presence time on sub-areas is proportional to sub-area size.

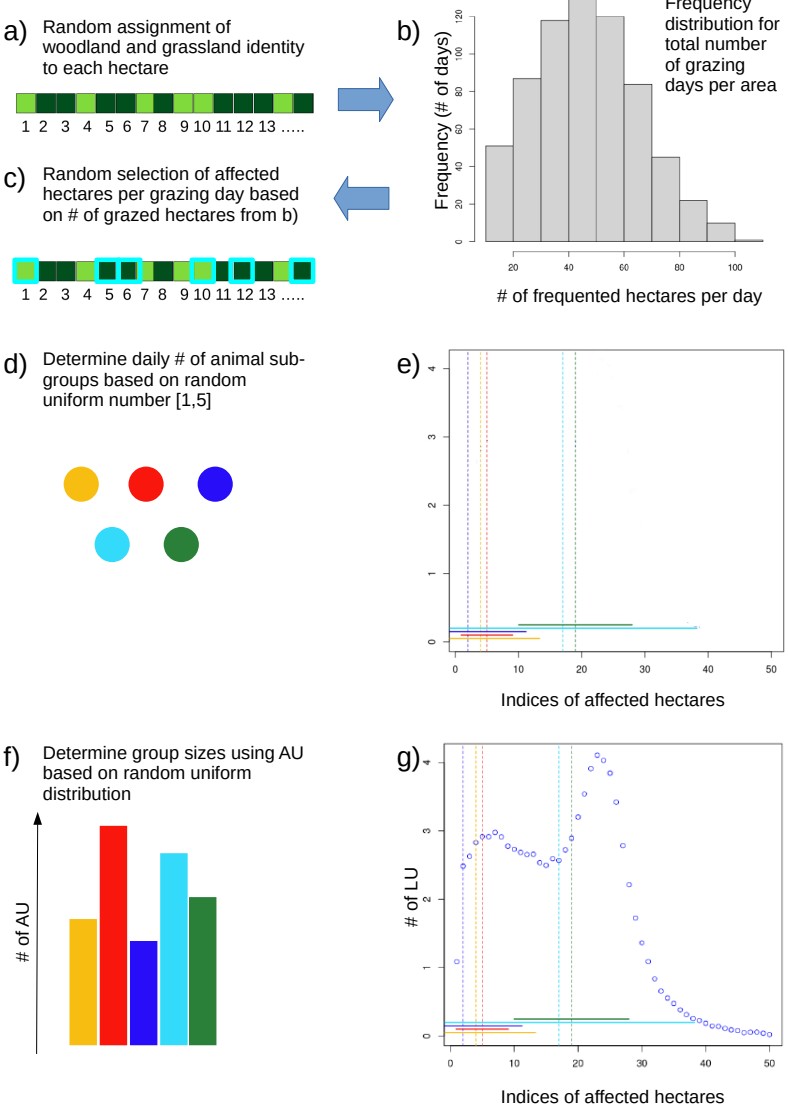

**Figure 3.** Illustration of the steps involved in creating the hectare-specific grazing sequences.

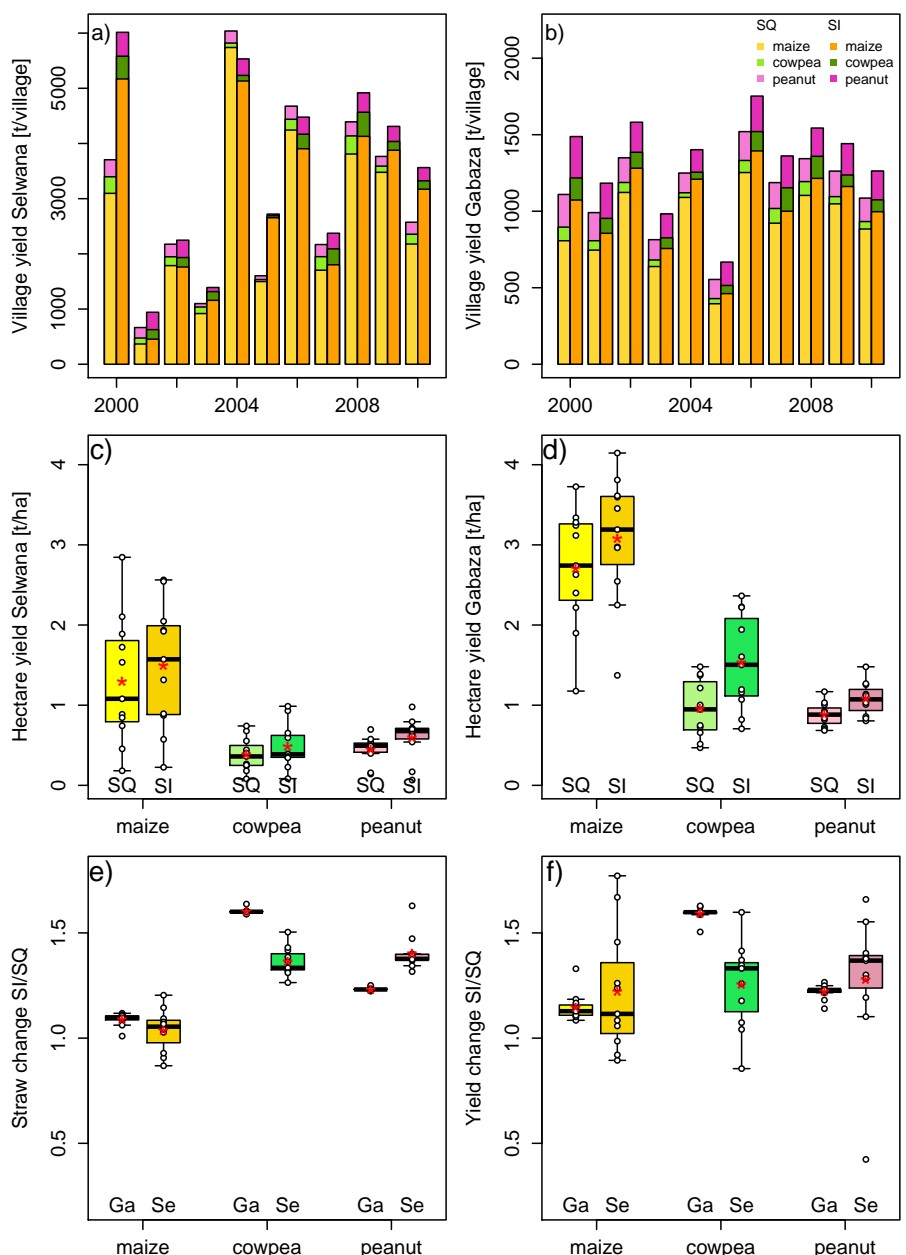

**Figure 4.** Time series of village-scale crop grain yields for Gabaza (panel a) and Selwana (panel b) under Status Quo management (SQ) and sustainable intensification (SI), and hectare-specific grain yields at Gabaza (panel c) and Selwana (panel d). Panels e) and f) summarize the change in grain yield and straw quantity between SI- and SQ-scenario at Gabaza (Ga) and Selwana (Se) for maize, cowpea and peanut, respectively. White dots in panels c) to f) depict the values of individual years, red asterisks the mean value.

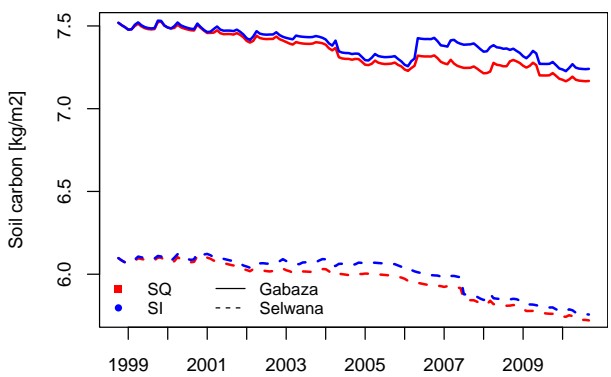

**Figure 5.** Time series of cropland soil organic matter (SOC) content at Gabaza and Selwana.

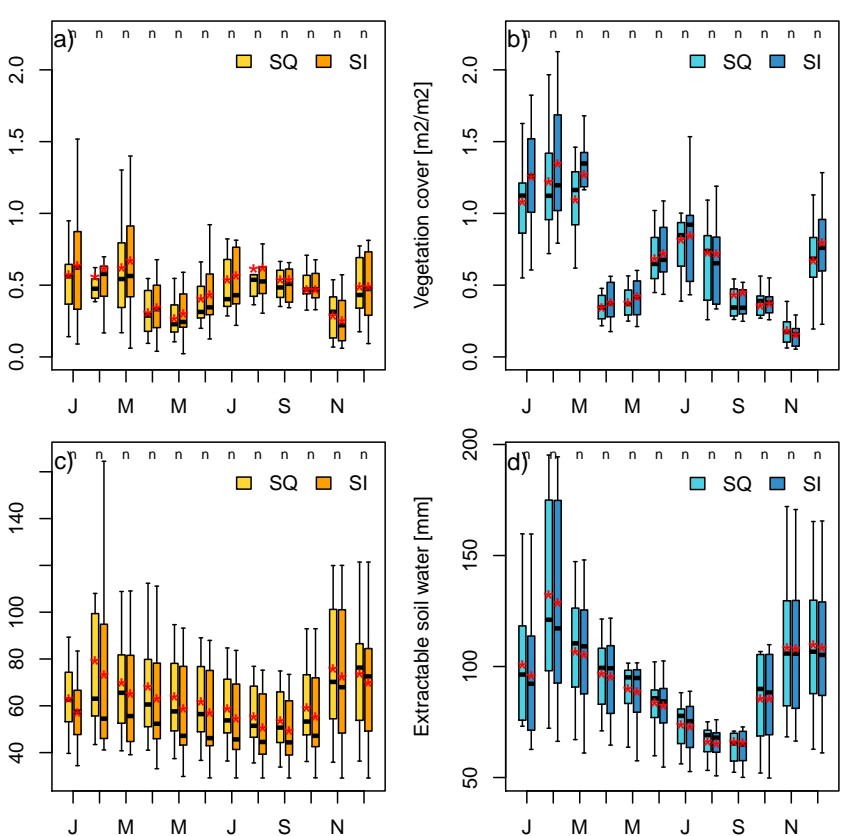

**Figure 6.** Monthly cropland leaf area index (LAI) at Selwana (panel a) and Gabaza (panel b), and monthly extractable soil water at Selwana (panel c) and Gabaza (panel d). SQ: Status quo management; SI: minimum sustainable intensification; n: difference non-significant between SQ and SI. Red asterisks: mean values.

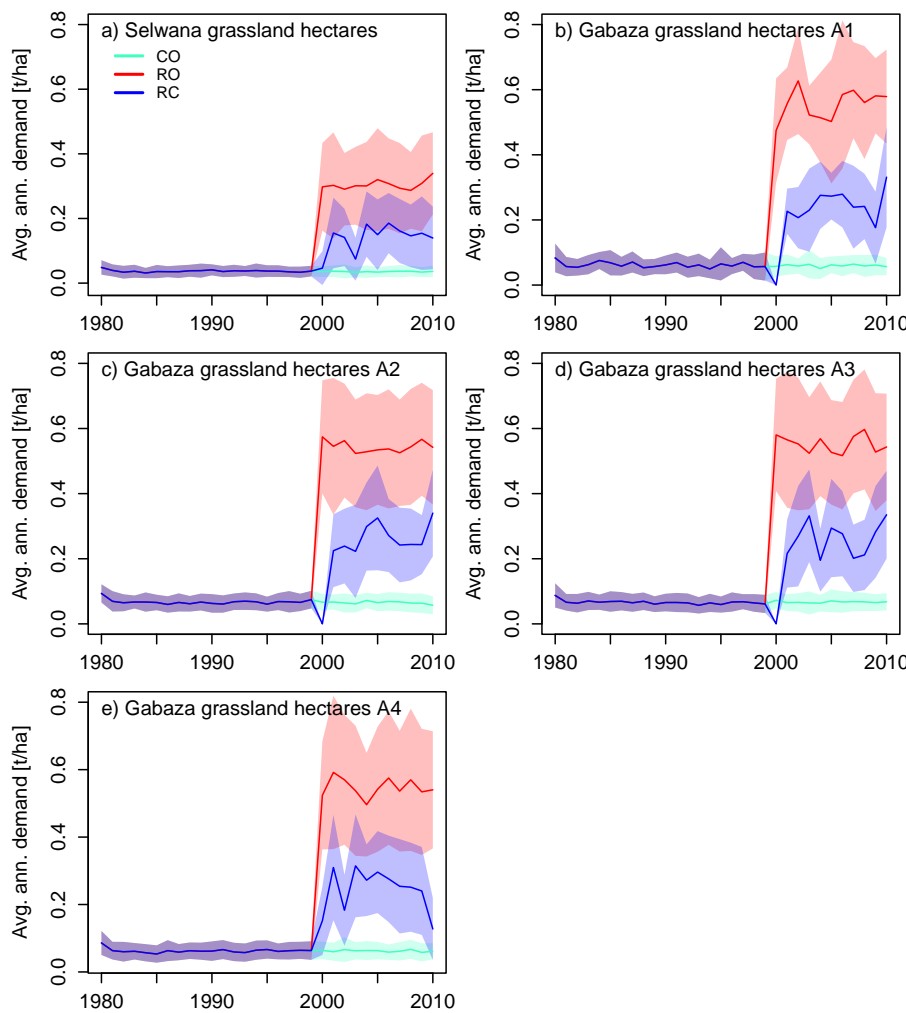

**Figure 7.** Average per-hectare annual biomass demand across simulated grassland hectares at Selwana (panel a) and the four sub-areas at Gabaza (panel b-e). Non-zero biomass demand in the CO-scenario is due to the prescribed low-intensity background grazing caused by small game when cattle are completely excluded.

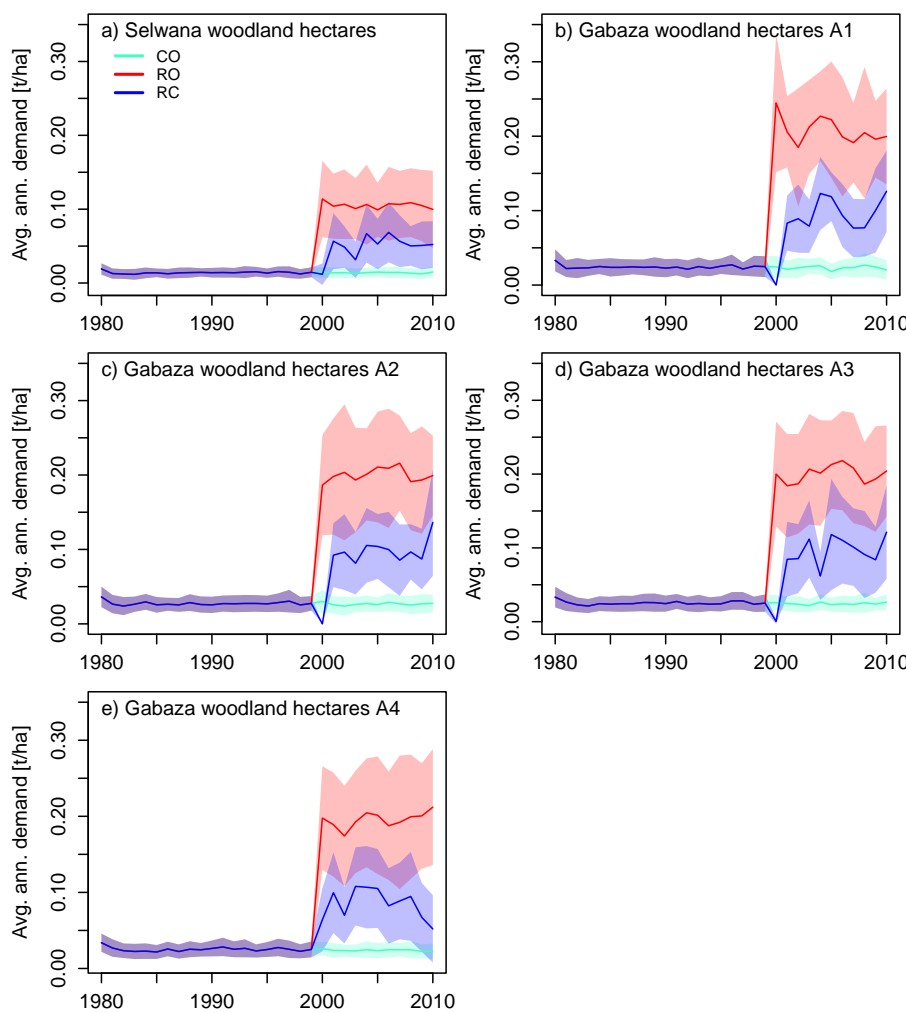

**Figure 8.** Average per-hectare annual biomass demand across simulated woodland hectares at Selwana (panel a) and the four sub-areas at Gabaza (panel b-e). Non-zero biomass demand in the CO-scenario is due to the prescribed low-intensity background grazing caused by small game when cattle are completely excluded.

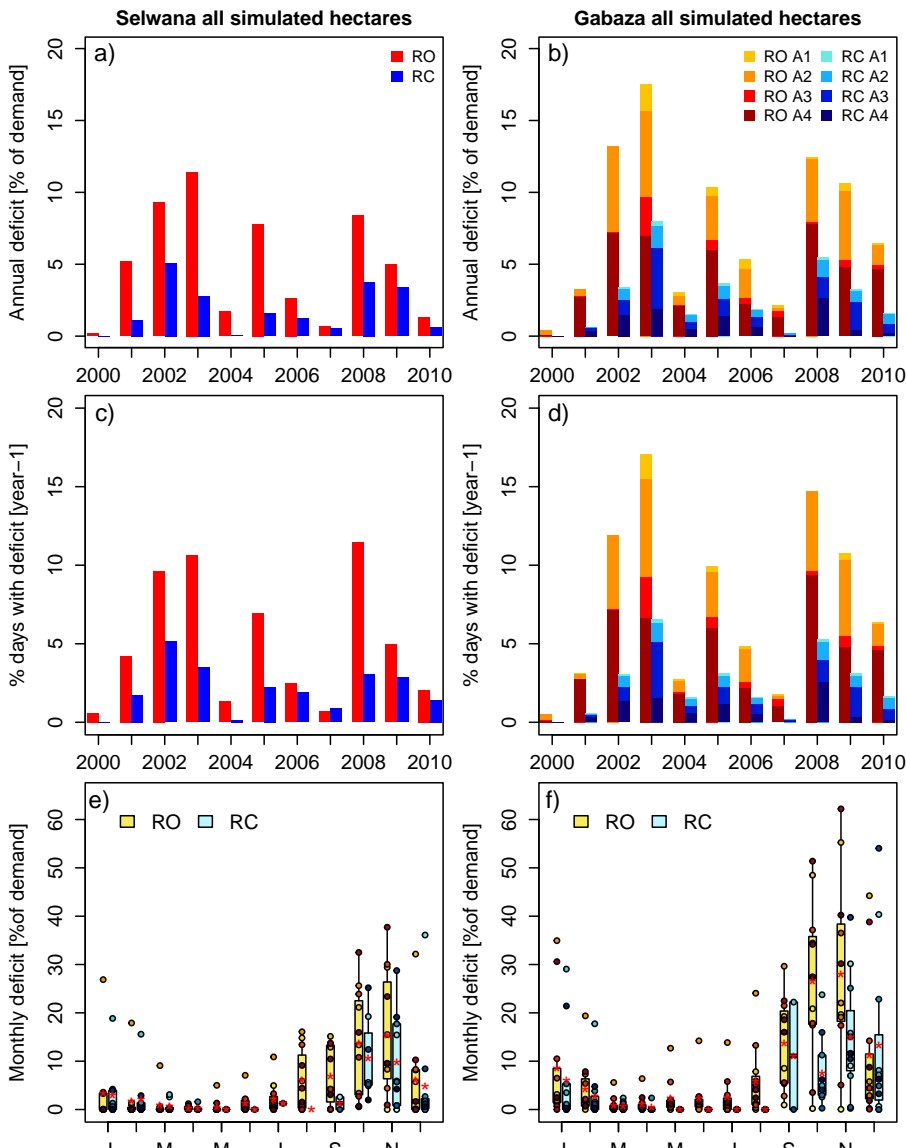

**Figure 9.** Demand-specific annual deficit across all simulated hectares, and percentage of grazing days integrated across all simulated hectares that had a deficit (irrespective of the size of the deficit), relative to the total grazing days within a year. Panel a) shows relative deficits for Selwana, panel b) shows relative deficits for Gabaza; Panel c) shows percentage of grazing days with deficit for Selwana, panel d) shows percentage of grazing days for Gabaza. Subdivisions of bars in panel b) and d) indicate the relative contribution of each sub-area to the site-scale annual deficit and days with deficit, respectively. Panels e) and f) show the monthly demand-specific deficits across all simulated hectares per village for Gabaza and Selwana, respectively. Red asterisks: mean values; White dots: annual values.

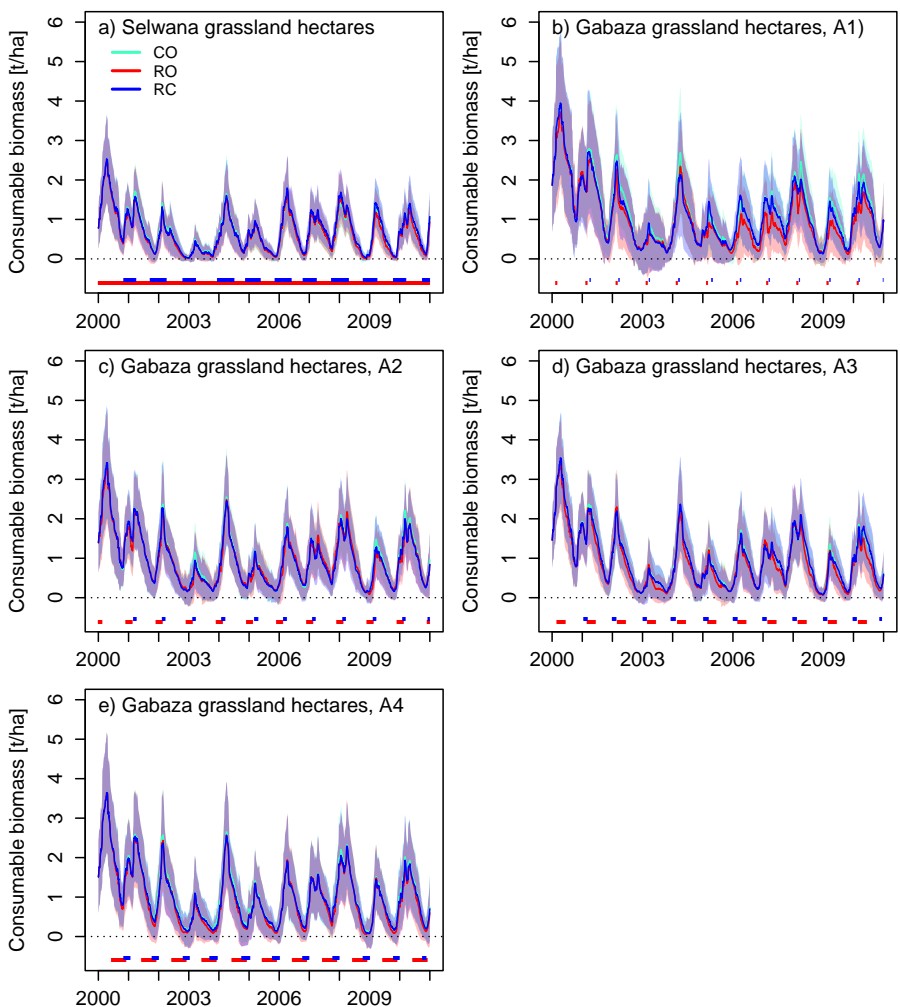

**Figure 10.** Temporal dynamics of average consumable grass biomass on simulated grassland hectares (living+dead standing grass leaf biomass+reproductive biomass, reduced by the minimum amount that is not available to grazers, i.e., 0.3 t/h for living and dead grass biomass, respectively, and 0.1 t/ha of reproductive biomass). Lines denote the mean across all simulated hectares, shaded areas show the standard deviation. The colored horizontal lines denote the animal presence times for the RO- and RC-scenario.

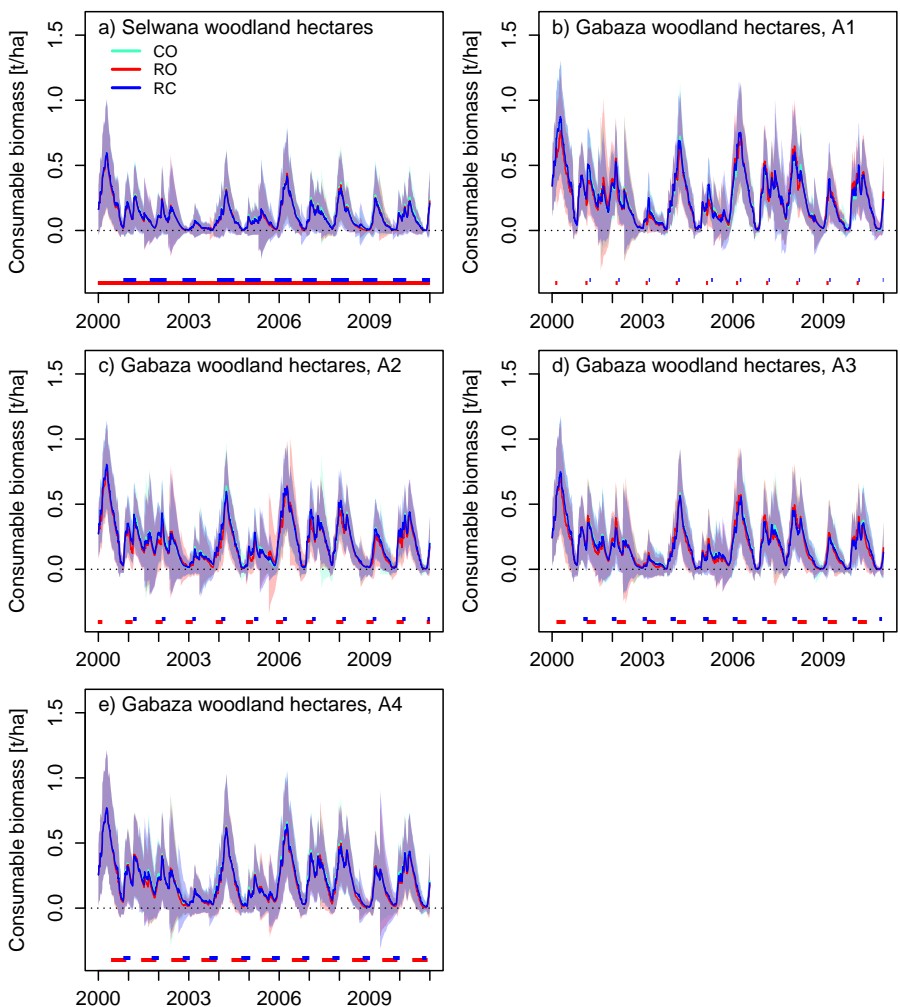

**Figure 11.** Temporal dynamics of average consumable grass biomass on simulated woodland hectares (living+dead standing grass leaf biomass+reproductive biomass, reduced by the minimum amount that is not available to grazers, i.e., 0.3 t/h for living and dead grass biomass, respectively, and 0.1 t/ha of reproductive biomass). Lines denote the mean across all simulated hectares, shaded areas the standard deviation. The horizontal lines at the bottom of the panels denote the respective animal presence times for the RO- and RC-scenario.

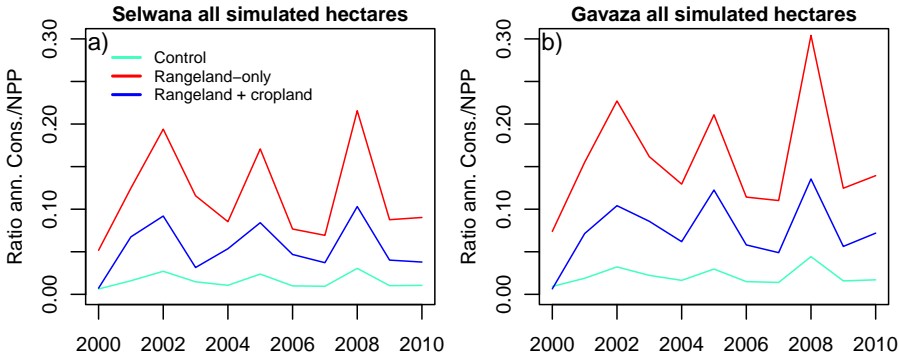

**Figure 12.** Temporal dynamics of consumption/NPP ratio, integrated across all simulated hectares per site.