# Peer review of "Effects of alternative crop-livestock management scenarios on selected ecosystem services in smallholder farming - a landscape perspective"

_Biogeosciences, 2022_

## Author Comment (AC1)

**RESPONSE TO REVIEWER 1 (CARSTEN MAHRON)**

Thank you very much for taking in the time to review our manuscript and your detailed comments and evaluation of our research.

**General comments**

The paper addresses relevant scientific questions within the scope of BG, here plant-animal-soil interactions between rangelands and cropping areas. Management strategies derived from scientific results are much needed as conflicts between farmers and herders are increasing in many savannahs. Novel concepts are presented, in this case, the coupling of a plot-scale agronomic plant-soil model with a landscape-scale vegetation model. The paper is written fluently and is well structured.

Thank you for your positive evaluation of our research.

As the model has not been calibrated and validated on-site and given that there are few significant differences between modeled scenario outputs, the innovations (coupling, landscape-scale, rangelands) and their added value for the model should be emphasized. This information is given, but rather hidden in Supplement B. Grazing rules are the linkage between the two (crop and vegetation) models and should be prominently explained in the Methods section (otherwise this info is missing to understand the feed gap section). Supplement B (walking distances etc) also explains the spatially explicit nature of the model, which is necessary for the landscape aspect highlighted in the title. As scenarios are hypothetical, they should have been chosen such that differences in model outcomes are clearer.

We will include a shortened version of Supplement B in the methods section to highlight both the grazing rules and the spatially explicit nature of the modeling approach. You are to some degree correct about the choice of scenarios with regard to the differences between model outcomes. In particular, for the APSIM scenarios, the sustainable intensification scenario defined very conservative improvements relative to the "business-as-usual" baseline scenario, which in turn caused moderate effect sizes. However, given that we wanted to test scenarios that can realistically be implemented by the smallholder farmers in our study villages as opposed to hypothetical scenarios chosen for maximum effect size, we decided against an SI-scenario that might have delivered substantial differences/improvements compared to the baseline but requires economic capacities that are not feasible for most of the local farmers. We will emphasize this incentive for our choice of scenario more strongly when introducing the scenarios in the methods section and add some supportive literature for our choice of the SI scenario (e.g. Nelson et al., 2022).

William C D Nelson et al., 2022. Tackling climate risk to sustainably intensify smallholder maize farming systems in southern Africa. Environ. Res. Lett. 17 (2022) 075005 https://doi.org/10.1088/1748-9326/ac77a3

The inclusion of fire events further blurs results, so that effects can be less clearly attributed to certain management.

The inclusion of fire in rangeland simulations was necessary to adequately represent the average vegetation dynamics and composition (tree-grass balance) of the savanna rangelands, as fire exclusion from simulations would lead to an expansion of woody vegetation with an associated loss of grass cover and biomass. Replicate simulations with the same ID were initialized with the same random seed between treatments and therefore have identical fire occurrence and vegetation dynamics during spin-up and up to the start of the grazing treatments, but start to deviate with the onset of differing grazing treatments between scenarios (as the grazing routine also requires the random number generator and therefore alters the sequence between different scenarios). To eliminate the effect of deviating fire history on biomass dynamics during the treatment phase, we additionally calculated the averaged timelines across replicates with the same fire

history by only considering those replicates that had no fire up to each given point in time of the treatment phase (Figs. S7, S8). In this case, grazing treatment is the only factor influencing differences between the three grazing scenarios.

**Specific comments**

**Title and Abstract:**

**Should the Title not indicate more clearly that this is a modelling study?**

Thanks for your suggestion, which led us to reconsider. We will change the title to "Modeling the effects of alternative crop-livestock management scenarios on important ecosystem services for smallholder farming from a landscape perspective".

The Abstract should highlight some of the findings related to ESF / ESS and landscape level, which are emphasized in the title

We will include your suggestion in the final version of the manuscript. We will also adjust the manuscript title to emphasize more clearly that we do not focus on all potential ESF / ESS at the landscape level, but on those that are most relevant for smallholder farming and that we can quantify with our modeling approach.

**Line 9: Please describe the current management in a few words (crop-livestock); do scenarios (ii) and (iii) include grazing access to croplands?**

Current management: minimum input crop-livestock agriculture => we will add this as a short description; Scenario (ii), i.e., the SI-scenario, does include cropland grazing. Scenario (iii) (rangeland-only scenario) does not allow cattle access to cropland at any time to provide a baseline for worst-case grazing deficits. Both cropland simulation scenarios (scenarios (i) and (ii)) included cattle access during the dry season. We will briefly clarify this at the mentioned place in the abstract.

**"Dry-season crop residue grazing substantially reduced feed deficits" – which scenario does this refer to?**

This refers to the RC-scenario, i.e., the rangeland grazing scenario where cattle had access to crop residues during the dry season, as opposed to the RO-scenario where cattle had no access to cropland at any time. We will clarify this point.

**Were "targeted irrigation" and "off-field residue feeding" tested in the scenarios? If not, it might be better to describe the model outcomes in more detail instead.**

Both aspects were not included in this modeling case study, but should be considered in further studies as we identified them as suitable candidates to further improve the supply of ESS at the landscape scale, beyond the measures identified directly in our study. We will make it more clear that these two recommendations are based on our expert assumptions, but were not part of the modeling study.

How are the impacts on "selected ecosystem services", implied by the title, represented in the abstract? Are yields and fodder supply seen as ESS? How about soil *C*, and soil water contents, which are also shown in the results section?

All of your listed indicators (yields, fodder supply, soil C, soil water content) are seen as ESS that can be quantified by the two models. We are aware that ESS comprise much more than these aspects and that there are also ecosystem services that we cannot quantify with the models; however, we have tried to include some

of the most important ones for smallholder farming – therefore we chose the phrasing "important ecosystem services in smallholder farming". We will add a short list of the ESS that we explicitly quantify.

**Introduction:**

A number of ecosystem functions and services are mentioned here (as would be expected from the title), but these are not reflected by / discussed in context with the model scenarios. E.g. run-off, soil erosion, evaporation, species diversity.

We will mention specifically which ESF and ESS we address in the context of this study, and which ones are relevant but not directly covered by the analysis of the model results.

How are ESS and ESF defined in this paper? Are agronomic measures (yield, biomass, LAI) also considered ESS / ESF?

Yield and biomass are provided by nature and benefit farmers and their livelihoods. We therefore consider them as ESS (see, e.g. Costanza et al 1997; MA, 2005) . LAI provides soil cover, protects against erosion, and directly links photosynthesis of leaves with the photosynthesis at the stand level, thus can serve as a measure to quantify ESF. We will define more precisely what we define as ESS and ESF in this paper, with a short reason why.

Costanza, R., d'Arge, R., de Groot, R. et al. The value of the world's ecosystem services and natural capital. *Nature* 387, 253–260 (1997). https://doi.org/10.1038/387253a0

MA (Millennium Ecosystem Assessment), 2005. Ecosystems and Human Well-being: Synthesis. Island Press, Washington, DC.

It is explained that APSIM represents the croplands and the DGVM the rangelands, but how are the animals represented? Is livestock represented in a process-based manner, i.e., including feedback between fodder and herd body weight, plant and dung quality etc. as in the cited LIVSIM studies?

Currently, livestock in aDGVM2 is represented as a non-interactive agent, i.e., the impact of livestock on vegetation is simulated, but the reciprocal effect on livestock is not integrated yet. This implies that aDGVM2 in its current state of development cannot simulate herd-related aspects such as animal growth, metabolism, reproduction, or nutrition status. Coupling aDGVM2 with a livestock model such as LIVSIM or an agent-based model such as RaMDry (Fust & Schlecht, 2018) would provide a way to also model animal-related aspects in a more process-based manner. We will clarify this point in the corresponding section of the introduction.

*Fust, Pascal and Eva Schlecht. "Integrating spatio-temporal variation in resource availability and herbivore movements into rangeland management: RaMDry—An agent-based model on livestock feeding ecology in a dynamic, heterogeneous, semi-arid environment." Ecological Modelling 369 (2018): 13-41.*

**Materials and Methods**

How did the two models interact, where they dynamically coupled (if so, how?), was a wrapper used or data "manually" transferred between models? At which intervals were data exchanged? What is the time step of each model?

The models were coupled offline. We used the same environmental input data to drive both models. APSIM simulations were conducted prior to the aDGVM2 simulations to determine (1) crop residue quantities available for livestock during the dry season, and (2) the timing of sowing and harvest to establish the times when cattle could access cropland in the SI and RC scenario and find sufficient feed. The cropland grazing time windows established through the APSIM simulations were then used to exclude livestock from rangeland in aDGVM2 (no grazing in aDGVM2 simulated during these time windows) in the RC-scenario. Both APSIM and aDGVM2 conduct simulations on a daily time step. We will add a brief explanation of this topic in this section.

**Does the SQ scenario include manure management, e.g. dung collection or corralling?**

Dung input to cropland during times of cropland grazing was considered in APSIM simulations. Dung collection on rangeland and transfer of dung from rangeland to cropland was not explicitly considered, as the style of cattle management on rangeland in the villages makes such an assumption very unlikely. Corralling was not considered in the simulations and is restricted to nighttime only at Gabaza, but not applicable at Selwana. We will clarify dung management in more detail.

**Section 2.3 explains that grazing management differs between the two surveyed villages. In how far does this affect the scenarios?**

First, the stocking density and therefore grazing intensity differs between both villages, which is directly accounted for in the aDGVM2 simulations via the daily demand. Second, the spatio-temporal distribution of grazing pressure differs between both villages, as the rangeland in Gabaza is sub-divided into four spatially separated sub-areas, whereas the rangeland at Selwana is one contiguous area that is continuously grazed during the time of cattle presence. This is also accounted for in the aDGVM2 simulations, where we explicitly simulate grazing on sub-areas in Gabaza by subdividing the overall grazing time window into four periods (see description of grazing routine in Supplementary part B). We could not account for differing daytime-night-time livestock handling between villages, as the temporal resolution of the grazing module in aDGVM2 is daily, not sub-daily. We will clarify which aspects of the village-specific grazing management measures were considered in the simulations.

**In the SI scenario, which species are used for rotation?**

In the selected SI scenario we apply the commonly practiced crop rotation where the staple crop maize is followed by a legume (usually peanut or cowpea). APSIM has been tested for such crop rotations in the region (see, Hoffmann et al 2020). Additionally, for the drier sites, we include a fallow period. The frequency of a crop within the rotation was determined by the land allocation of the crop observed in the field.

Hoffmann, M. P., Swanepoel, C. M., Nelson, W. C. D., Beukes, D. J., van der Laan, M., Hargreaves, J. N. G., Rötter, R. P. (2020). Simulating medium-term effects of cropping system diversification on soil fertility and

crop productivity in southern Africa European Journal of Agronomy 119, 126089, https://doi.org/10.1016/j.eja.2020.126089

In Table 1, it would be good to calculate stocking density.

We will add it.

**Line 128: Was feed demand (parameterised as) constant over time?**

Yes. This assumption is a simplification we made as aDGVM2 can simulate biomass quantities, but not quantify changes in biomass quality at this point. We will make this clear in the manuscript.

**Line 144: Figs 1e, 1f and 2 don't show when crops are harvested, but when livestock is present.**

Strictly speaking, this is correct. However, indirectly the figures also show the timing of crop sowing and harvest, because (with a buffer of two weeks between harvest and livestock arrival / livestock leaving and sowing) the times of animal absence are the times of crop cultivation. We will clarify this in our phrasing.

*Line 170: Grazing on random days – were these days the same for all model runs or was a probabilistic approach taken? Information from Supplement B should be shown here (grazing rules).*

Supplement B will be moved from Supplement and incorporated into the Materials and Methods section

**Line 175: Were fire events the same for all model runs?**

Each individual replicate simulation within a scenario was initialized with a different random seed. Therefore, the sequence of fire occurrence differed between replicate simulations, i.e., each individual simulated hectare had its own fire sequence, implicitly implying that fires were small enough to not fully burn an entire grazing area at a given time. This reflects the commonly observed fire regime in the region (predominance of low-intensity grass layer fires). However, we used the same set of replicates for all three scenarios, implying that the fire event sequences between scenarios did not differ up to the point where the grazing treatments started. As the grazing module also uses random numbers, differing grazing regimes between scenarios implied diverging random number sequences upon the start of the grazing treatments, and therefore automatically a deviation of fire events during the treatment period. As grazing and fire also interact in the field, e.g., via effects of grazing on fuel availability, we did not see this as a problem, although it can make the evaluation of treatment effects more difficult. (but still possible, see, e.g., Figs. S7 and S8 as examples for the elimination of fire effects in treatment evaluation).

Line 195 ff. not clear

We will expand this sub-section by moving part of supplement B in this place to make it more clear.

*Line 215: Descriptions in this section are somewhat difficult to follow, would be good to refer to a figure showing the resp. outputs (Fig 6?).*

We do not deem it good practice to refer to a result figure in the methods section, where we are not yet presenting and explaining the results shown in the figure. This might cause additional unclarity/confusion for the reader. We will consider adding a conceptual figure illustrating the workflow.

**Results**

Has APSIM been calibrated / calidated regarding crop yields? The maize yields for Gabaza appear relatively high (also compared to what has been stated in the methods section).

Our study is built on the basis of more detailed specific investigations and evaluations of modeling growth and development of various field crops /cropping systems in the region using APSIM. We have calibrated and validated the APSIM model for sites and target crops in the study region, as reported, e.g., in Hoffmann et al. (2018; 2020) and Nelson et al (2022). The evaluations include a wide range of crops and crop rotations; Hoffmann et al. (2018) looked specifically at peanut. In Nelson et al. (2022), we focused on the calibration of maize for the study sites (5 villages in the Mopani district in Limpopo province, South Africa); a number of complex crop rotations were tested at sites in South Africa by Hoffmann et al 2020). We compared surveyed yields against simulated maize yields (see Figure 4 from Nelson et al. (2022)). Surveyed results from 2019 are highly variable starting from below 500 kg/ha up to almost 3000 kg/ha in Gabaza.

**Hoffmann, M.P.**, Swanepoel C.M., Nelson, W.C.D., Beukes, D.J., van der Laan, M., Hargreaves, J.N.G., Roetter, R.P. (2020). Simulating medium-term effects of cropping system diversification on soil fertility and crop productivity in southern Africa. European Journal of Agronomy 126089. doi.org/10.1016/j.eja.2020.126089

*Hoffmann, M.P.*, Odhiambo, J.J.O., Koch, M., Ayisi, K.K., Zhao, G., Soler, A.S., Roetter, R.P. (2018) Exploring adaptations of groundnut cropping to prevailing climate variability and extremes in Limpopo Province, South Africa. Field Crops Research 219: 1-13. doi: 10.1016/j.fcr.2018.01.019

Fig. 3a and b: The stacked bars make visual comparison for peanut and cowpea between SQ and SI scenarios relatively difficult. One chart per crop, in parallel to the description in the text, might facilitate interpretation.

By making the stacked bar plots, we had decided to condense the visualization of the results to save space. We will rearrange these results in separate panels according to your suggestion, and add them as a separate figure.

Line 258: The statement "For cowpea and peanut, SI had a stronger positive effect at Gabaza for relative and hectare-specific increases." seems to contradict the numbers presented for peanut (factor 1.22 in Gabaza and 1.28 in Selwana).

You are correct, this got mixed up. The SI effect was only stronger for cowpea, not for peanuts. We will correct this.

Line 271: "SI reduced SOC-loss to 3.70%, [...]" should probably say "by 3.70%".

No, not "by". SI reduced SOC loss from 4.68% in the SQ-scenario to a loss of 3.70%. We will rephrase this to make it more clear how we mean it.

Section 3.2.1: The term biomass could be changed into pasture to avoid confusion with crop residue consumption.

Animals consume biomass on cropland as well as on pasture, but the quality of the biomass differs (crop biomass vs. grass biomass). We will change "biomass" to "grass biomass" in this section to avoid confusion.

Table 2a and the respective description in the text are very hard to read; why not showing Fig S3 and S4 instead and moving Table 2a to the supplements (for those who are interested in the exact numbers)? The text could be limited to the main trends and comparisons instead of repeating means and standard deviations from the table.

We will change this section according to your suggestion.

**Line 300: Why was the number of animals lower in woodlands compared to grasslands?**

The explanation for this is given in Supplement B lines 51-53: "When assigning the AU to the affected hectares, we randomly assigned the lower range of AU numbers to the woodland and the higher quantities of AU to the grassland hectares to consider the higher feed availability on grassland than woodland." I.e., this is due to the way we distributed the animal numbers across the individual hectares selected for grazing on a given day. As explained in supplement B, we first draw the number of grazing-affected hectares, then split the animal number according to the described scheme to decide how many animals each affected hectare has to supply on a given day, and in a final step have to assign these animal numbers per hectare to specific hectares. For this step, we sort the number of animals per hectare in descending order and assign the lower

range of the ordered list to those affected hectares that are woodland hectares. Grass biomass availability on woodland hectares is lower than on grassland hectares, therefore a woodland hectare can supply fewer animals than a grassland hectare. We assume that animals will preferentially go to places of higher feed availability and therefore deem our decision to distribute animals across hectares in this fashion as reasonable. An agent-based cattle model would allow animals to interactively decide where to go based on where biomass is available, but due to our lack of such an animal scheme, we have to mimic it artificially. We will expand our explanation accordingly and add it to the methods section together when merging Supplement B into it.

Section 3.2.2: The feed deficit at Gabana (Fig 6 b and d) raises the question on grazing decisions: Were these decisions dynamic (taken by the model during the run) and animals moved to another grazing area when pasture became limiting, or were grazing periods per area determined before the model run started? This should be explained in the methods section.

The grazing sequences were established offline, i.e., prior to the aDGVM2 simulation runs, for the entire simulation period and used as input to the vegetation model. For Gabaza, we split the overall animal presence time on rangeland into four sub-periods such that presence durations were proportional to the size of each sub-area. We actually already explain this in subsection 2.6.3, in lines 190 ff.: "Additionally, we partitioned presence time proportionally to sub-area size. Therefore, the average total annual amount of dry matter removed from a given hectare is approximately constant, independent of its location

in a small or large sub-area, but resting periods are longer for the smaller sub-areas." We will rephrase this part to make the explanation more prominent/clear.

Secondly, some areas in Gabana were affected more seriously (frequently) by feed gaps (RO A2 and A4, RC A3). Was this because animals stayed there longer or because the areas were smaller or due to the timing of grazing within a season?

By splitting animal presence duration on each sub-area proportionally to area size (see preceding comment), the average annual grazing load per hectare is equal for all sub-areas, independent of their size. Therefore, area size should not matter and feed gap differences between sub-areas are likely attributable to the timing of grazing within a season.

Line 347f. "grazing frequently caused significantly (two-sided t-test with p < 0.05) higher average biomassnormalized GPP and NPP values relative to control" – was there an optimum grazing frequency for pasture regeneration?

We suspect that an optimum grazing frequency exists where simulated biomass-normalized productivity is maximized, but we did not look into this question in the context of this study. Despite the seasonally occurring grazing deficits, from a vegetation perspective, the grazing intensities simulated in this study are low to moderate, and based on biomass comparison between control and grazing scenarios overgrazing was not a problem.

**Discussion**

**Management-related differences between villages could be discussed in more depth? Effects of SI between sites (expected to be stronger on the more extensive = poorer site)?**

Predicting in which village SI will have a stronger effect is challenging due to the variety of influencing factors that may enforce, but in most cases counter SI effects. We expected that relative improvement compared to Status Quo should be stronger at Selwana where SQ produced lower yields per hectare compared to Selwana and therefore the potential for improvement seemed higher. However, although relative yield increases at Selwana were indeed higher in good years, more pronounced water availability constraints under SI led to small increases or even yield decreases in some years. Therefore, the response variability was greater at Selwana compared to Gabaza. Moreover, within-crop-type response intensity was not consistent between sites, with cowpea showing the strongest relative response at Gabaza. We will add a short discussion on this topic in section 4.1.

**Did more frequent grazing on certain areas affect the regeneration of pasture (positively or negatively)?**

The effect of grazing intensity/frequency on pasture regeneration was not investigated in this study. However, we focus on pasture regeneration in the context of drought and grazing in another study that is currently in preparation (Behn et al., in prep).

Line 376: "SI-measures could result in yield losses in dry years due to enhanced crop growth and associated increased water demand" – yield loss due to enhanced crop growth sounds paradoxical. It is explained in lines 386 ff and I would suggest to move this sentence there.

We will move the explanation as suggested.

**Line 385: Not sure whether N input would increase cowpea growth.**

True. In the case of cowpea (and peanut) water limitation is likely the stronger growth constraint than nitrogen limitation, although enhanced symbiotic nitrogen fixation is also associated with a cost to the plant that can reduce growth compared to a well-supplied legume. In addition, other nutrients (phosphorus, potassium) may still be limiting without fertilization. We will address this point in more detail in the final manuscript.

*Line* 396: *Loosening sandy soils and increasing infiltration to increase plant growth – these measures appear to be more appropriate for heavy soils.*

Indeed, this management measure may be less applicable for soils with very high sand contents. However, the sandy clay loam at Gabaza and sandy loam at Selwana may benefit from loosening where the clay and silt components make the substrate prone to slake, experience hardsetting, and building of surface crusts, which results in poor water and air infiltration and can increase erosion risk. We will add this detail to the discussion at the specified location in the text.

Lines 408 ff.: Could undergrazing be a problem (grass becoming moribund)? Does the model account for stimulated regrowth by grazing? Is pasture quality considered in the model? This is mentioned later (lines 454ff.), perhaps both paragraphs could be better connected.

We will move the corresponding section from lines 454ff to be integrated into the section in lines 408 ff.

**Line 423: Sounds unlikely that vegetation growth reacts with 2-3 months delay to the onset of rains.**

This is poorly phrased. Vegetation growth starts shortly after the onset of the rains, but peak biomass is reached with a delay of 2-3 months after the beginning of the wet season. During the time when biomass ingrowth takes place, quantities a) are not yet sufficient to fully supply the demand and b) grazing can additionally slow the development. We will explain this more clearly in the revised manuscript.

Section 4.3: Surprising that farmers do not store crop residues; this is common practice from Senegal to Ethiopia in densely populated areas. In this context, population density in the research area might be interesting (in the MatMet section).

Thank you for pointing this out, we will add it to the discussion.

Line 444f.: Pasture quantity is only part of the problem, correct, and so is pasture quality (high lignin contents, if moribund biomass or standing litter are fed).

Correct, quality is also important, although it is currently only crudely accounted for in the aDGVM2 grazing scheme, where dead biomass is assigned 3/3 of the nutrition content of living grass biomass. We will add this point to the discussion.

*Lines 451 ff.: Did the decreasing trend in SOC stem from APSIM or from the DGVM (or both)? Should the control treatment not represent (and the models be calibrated to) carbon equilibrium?*

The reported SOM-trend is for cropland only, i.e., originates from APSIM (for details of handling SOC in APSIM, see Hoffmann et al 2020). We do track soil carbon in aDGVM2, but did not analyze the results because SOM dynamics in the version we used for the study simulations has not yet been benchmarked.

In context with line 258, stronger effect of SI at Gabaza (one would expect higher impact in the poorer environment, i.e. in Selwana) could be discussed.

See our reply to your first comment for the discussion section.

**Conclusions**

The section summarizes added value of the coupling well and identifies the next steps (inclusion of an animal and a herder decision model).

Thank you for this positive acknowledgment.

*Line* 468: *Holistic management is a much disputed strategy that should probably not be introduced in the last section without further explanation. What is meant here is probably integrated crop-livestock systems.*

We will rephrase accordingly.

**Supplements**

Fig S1: As bars are the same and only the y-axis label differs, one of both subfigures could be omitted.

We will adjust the figure according to your suggestion and merge both panels by adding a second y-axis on the right-hand side of the first panel.

**Fig S2: Differences between solid and hatched signature are not visible**

We will think about a better way to visualize the difference between SQ and SI scenario.

**S3 and S4: Why is biomass demand for CO (no animals) > 0?**

As mentioned in the methods section, we prescribed a very low demand for spinup and control to establish a grass community that is generally accommodated to grazing. Even without livestock, grazing by small game is common and creates some small background demand in the CO-scenario.

Fig.s 7 and following: What are the red and blue lines at the bottom (should be explained in the captions, so that the figures are self-explaining)? Why are standard deviations for CO sometimes (around 2007 in Gabaza A1 and A3) higher than for the grazed treatments?

Thank you for pointing this out, the explanation should indeed be part of the caption (it is the animal presence timelines for the two grazing scenarios). It is not the standard deviations in the CO-scenario that are higher (they are in the same range as for the two grazing scenarios), but actually, the mean of the CO-scenario that is higher than the mean of the two grazing scenarios at these times, and therefore also the enveloping standard deviation around the mean. In these years, the seasonal impact of the grazing led to a visible decrease of biomass in the grazing scenarios compared to the control.

---

## Author Comment (AC2)

RESPONSE TO REVIEWER 2 (ANONYMOUS)

*The present manuscript by Pfeiffer et al. is a comprehensive review of key factors involved in the sustainability and productivity of African smallholder farms. I found the analysis to be insightful and appropriate and only have minor comments.*

Thank you for your positive evaluation of your work.

*The abstract would benefit from more numbers to make it more quantitative to help describe exactly what was found.*

We will add some numbers for the key results.

*There are minor usage issues throughout, the first is on line 34 (p. 2): "Livestock" is plural so the correct usage is 'Livestock provide..' (see also line 35; a quick review will correct any minor issues.) More: space after the period on L. 70, etc. (L. 89: "and cowpea" to distinguish between tubers after the comma.)*

Thank you for these editorial hints, we will incorporate them in the final manuscript.

*I found the last paragraph of the Introduction to be a bit confusing: listing the questions first and then the approach used would help lead into the specific study.*

We will restructure the paragraph to improve the flow of the logic.

*section 2.2: some of the crops are listed twice in the paragraphs beginning lines 99 and 105.*

We have checked this and found it is not the case; there is also no duplication in table 1.

*Please describe t/ha to distinguish between metric tons (or tonnes) and imperial tons. Obviously the former is more appropriate and I assume is used here, but the latter is in common usage in many places.*

It is metric tons. We will specify this explicitly when introducing the unit for the first time, or alternatively exchange the unit tons with Mg.

*LU is not defined at first use.*

Livestock unit. Sorry for the omission, we will add the definition.

*Results: I feel that there are probably too many significant digits for a modeling study throughout. For example 53 +/- 23 is probably more correct than 53.2 +/- 22.9, etc. The results were comprehensive but somewhat long, and an eye toward brevity would improve the Results section.*

We will check the number of significant digits throughout the manuscript, and see if/where we can shorten the results section.

*I guess that my biggest question regarding the outcomes is the suggestion for irrigation feasibility? This usually involves considerable expense and can have other deleterious consequences. A brief analysis of the likelihood or sustainability of irrigation would strengthen the conclusions.*

In our project area, there have been recent investigations (e.g. Lam et al., in review) that have shown that in some of the water catchments in future, there will indeed be a decline in available surface water and groundwater resources. In our scenarios, however, we have been referring to very restricted "deficit irrigation" which can be drawn from rainwater-harvesting, and drip irrigation from available boreholes and surface water without exhausting water resources as reported in Magombeyi et al 2018;  Parry et al 2020).

*Magombeyi M S, Taigbenu A E and Barron J 2018 Effectiveness of agricultural water management technologies on rainfed cereals, crop yield and runoff in a semi-arid catchment: a meta-analysis J. Agric. Sustain. 16 418–41*

*Parry K, van Rooyen A F, Bjornlund H, Kissoly L, Moyo M and de Sousa W 2020 The importance of learning processes in transitioning small-scale irrigation schemes Int. J. Water Resour. Dev. 36 199–233*

---

## Author Response (AR1)

Dear editor,

we have now addressed the points raised by the two reviewers for our manuscript based on our suggestions that we provided for each list of reviewer comments after closure of the discussion phase. The required edits have been incorporated in the revised version of the manuscript and supplementary material, and are highlighted in blue (edits made in response to reviewer 1, Carsten Mahron) and red (edits made in response to reviewer 2). The following part states, for each comment, how and where we have addressed the raised topic in the revised version.

Kind regards on behalf of all authors,

Mirjam Pfeiffer

**Edits in response to reviewer 1 (Carsten Mahron)**

*As the model has not been calibrated and validated on-site and given that there are few significant differences between modeled scenario outputs, the innovations (coupling, landscape-scale, rangelands) and their added value for the model should be emphasized. This information is given, but rather hidden in Supplement B. Grazing rules are the linkage between the two (crop and vegetation) models and should be prominently explained in the Methods section (otherwise this info is missing to understand the feed gap section). Supplement B (walking distances etc) also explains the spatially explicit nature of the model, which is necessary for the landscape aspect highlighted in the title. As scenarios are hypothetical, they should have been chosen such that differences in model outcomes are clearer.*

We now include the slightly shortened information that was previously in Supplement B in the methods description (sections 2.7.4 and 2.7.5).

*The inclusion of fire events further blurs results, so that effects can be less clearly attributed to certain management.*

We have added some more information on the handling of fire in our simulations to clarify if and how fire occurrence differs between grazing scenarios and to give a justification why we had to include it nonetheless (section 2.7.1. lines 218-225).

*Should the Title not indicate more clearly that this is a modeling study?*

We have changed the title to "Modeling the effects of alternative crop-livestock management scenarios on important ecosystem services for smallholder farming from a landscape perspective", to make it clear that this is a modeling study that focuses on selected ecosystem services that are important for smallholder farmers.

*The Abstract should highlight some of the findings related to ESF / ESS and landscape level, which are emphasized in the title*

We now specifically name the ESS and ESF that were considered in this study (l. 12/13).

*Line 9: Please describe the current management in a few words (crop-livestock); do scenarios (ii) and (iii) include grazing access to croplands?*

We have added a brief description characterizing the current management ("minimum input crop-livestock agriculture", l. 9/10) and explicit statements which scenarios include or exclude cropland grazing (l. 10, l. 11).

*"Dry-season crop residue grazing substantially reduced feed deficits" – which scenario does this refer to?*

Reference is the rangeland-only scenario. We have added this information (l. 17).

*Were "targeted irrigation" and "off-field residue feeding" tested in the scenarios? If not, it might be better to describe the model outcomes in more detail instead.*

We now make it clear that targeted irrigation is a measure we expect to improve the situation (l. 18), and have added (also in response to the demand of the second reviewer to provide some numbers for the key results) more detailed quantitative information for the key outcomes.

*How are the impacts on "selected ecosystem services", implied by the title, represented in the abstract? Are yields and fodder supply seen as ESS? How about soil C, and soil water contents, which are also shown in the results section?*

All of the listed indicators (yields, fodder supply, soil C, soil water content) are seen as ESS that can be quantified by the two models. We are aware that ESS comprise much more than these aspects and that there are also ecosystem services that we cannot quantify with the models; however, we have tried to include some of the most important ones for smallholder farming. We have changed the title of our study to make it more clear that we focus on a subset of potential ESS that are of particular relevance for smallholder farmers in (Southern) Africa, and have added a short listing of the ESS that we explicitly quantify in this study (l. 12/13).

**Introduction**

*A number of ecosystem functions and services are mentioned here (as would be expected from the title), but these are not reflected by / discussed in context with the model scenarios. E.g. run-off, soil erosion, evaporation, species diversity.*

We now explicitly state which ESF and ESS we address in the context of this study, and which ones are relevant but not directly covered by the analysis of the model results (l. 89-94).

*How are ESS and ESF defined in this paper? Are agronomic measures (yield, biomass, LAI) also considered ESS / ESF?*

Yield and biomass are provided by nature and benefit farmers and their livelihoods. We therefore consider them as ESS (see, e.g. Costanza et al 1997; MA, 2005), see addition in lines 33-35 . LAI provides soil cover, protects against erosion, and directly links photosynthesis of leaves with the photosynthesis at the stand level, thus can serve as a measure to quantify ESF. We now define more precisely what we define as ESS and ESF in this paper, with a short reason why (l. 89-94).

*It is explained that APSIM represents the croplands and the DGVM the rangelands, but how are the animals represented? Is livestock represented in a process-based manner, i.e., including feedback between fodder and herd body weight, plant and dung quality etc. as in the cited LIVSIM studies?*

We have clarified this point in l. 70-73.

**Materials and Methods**

*How did the two models interact, where they dynamically coupled (if so, how?), was a wrapper used or data "manually" transferred between models? At which intervals were data exchanged? What is the time step of each model?*

We have added a short section (section 2.4, l. 161-167) that explains these details.

*Does the SQ scenario include manure management, e.g. dung collection or corralling?*

Dung input to cropland during times of cropland grazing was considered in APSIM simulations, but we did not simulate dung collection from corralling areas or rangeland (see addition in l. 134-137).

*Section 2.3 explains that grazing management differs between the two surveyed villages. In how far does this affect the scenarios?*

We have added information on this topic in lines 154-160.

*In the SI scenario, which species are used for rotation?*

We have added this information in lines 131-134.

*In Table 1, it would be good to calculate stocking density.*

We have added stocking density to Table 1.

*Line 128: Was feed demand (parameterised as) constant over time?*

Yes. This assumption is a simplification we made as aDGVM2 can simulate biomass quantities, but not quantify changes in biomass quality at this point. We will make this clear in the manuscript. We have made this more clear than it was before in lines 149/150.

*Line 144: Figs 1e, 1f and 2 don't show when crops are harvested, but when livestock is present.*

Strictly speaking, this is correct. However, indirectly the figures also show the timing of crop sowing and harvest, because (with a buffer of two weeks between harvest and livestock arrival / livestock leaving and sowing) the times of animal absence are the times of crop cultivation. We have clarified our phrasing in l. 185).

*Line 170: Grazing on random days – were these days the same for all model runs or was a probabilistic approach taken? Information from Supplement B should be shown here (grazing rules).*

Supplement B is now moved and incorporated into the Materials and Methods section (sections 2.7.4 and 2.7.5).

*Line 175: Were fire events the same for all model runs?*

We have added an explanation on the details of fire inclusion in l. 219-225.

This lack of clarity was related to information from the former supplement B that is now provided in sections 2.7.4 and 2.7.5.

*Line 215: Descriptions in this section are somewhat difficult to follow, would be good to refer to a figure showing the resp. outputs (Fig 6?).*

We do not deem it good practice to refer to a result figure in the methods section, where we are not yet presenting and explaining the results shown in the figure. We have rephrased the description of our workflow in lines 298-302 to be more concise and hopefully easier to understand than it was before.

**Results**

*Has APSIM been calibrated / calidated regarding crop yields? The maize yields for Gabaza appear relatively high (also compared to what has been stated in the methods section).*

We have added information on the calibration and validation of APSIM with regard to the study region in lines 175-179.

*Fig. 3a and b: The stacked bars make visual comparison for peanut and cowpea between SQ and SI scenarios relatively difficult. One chart per crop, in parallel to the description in the text, might facilitate interpretation.*

We have made a separate figure that shows one chart per crop and village, and added it to the supplementary material (Fig. S2, referenced in l. 317/318 of the main manuscript) as additional information for any readers how may be interested to see the detailed information, but decided to keep the original figure in the main manuscript to save space.

 *The statement "For cowpea and peanut, SI had a stronger positive effect at Gabaza for relative and hectare-specific increases." seems to contradict the numbers presented for peanut (factor 1.22 in Gabaza and 1.28 in Selwana).*

We have corrected this mistake.

 *"SI reduced SOC-loss to 3.70%, [...]" should probably say "by 3.70%".*

No, not "by". SI reduced SOC loss from 4.68% in the SQ-scenario to a loss of 3.70%. We have rephrased this misleading part of the sentence (l. 359).

*Section 3.2.1: The term biomass could be changed into pasture to avoid confusion with crop residue consumption.*

We have changed "biomass" to "grass biomass" in this section to avoid confusion.

*Table 2a and the respective description in the text are very hard to read; why not showing Fig S3 and S4 instead and moving Table 2a to the supplements (for those who are interested in the exact numbers)? The*

*text could be limited to the main trends and comparisons instead of repeating means and standard deviations from the table.*

We have changed this section according to the suggestion.

*Line 300: Why was the number of animals lower in woodlands compared to grasslands?*

The explanation for this is was originally provided in Supplement B and is now included in the Materials and Methods section (l. 280-282).

*Section 3.2.2: The feed deficit at Gabana (Fig 6 b and d) raises the question on grazing decisions: Were these decisions dynamic (taken by the model during the run) and animals moved to another grazing area when pasture became limiting, or were grazing periods per area determined before the model run started? This should be explained in the methods section.*

We explain this in subsection 2.7.3, in lines 190 ff.: "Additionally, we partitioned presence time proportionally to sub-area size." We have now added an explicit statement that due to this size-proportional time-split between sub-areas, the annual demand is approximately equal for all hectares, independent of their location in a specific sub-area (l. 239/240).

*Secondly, some areas in Gabana were affected more seriously (frequently) by feed gaps (RO A2 and A4, RC A3). Was this because animals stayed there longer or because the areas were smaller or due to the timing of grazing within a season?*

By splitting animal presence duration on each sub-area proportionally to area size (see preceding comment), the average annual grazing load per hectare is equal for all sub-areas, independent of their size. Therefore, area size should not matter and feed gap differences between sub-areas are attributable to the timing of grazing within a season. We have added a statement on this in l. 241-242.

*Line 347f. "grazing frequently caused significantly (two-sided t-test with p<0.05) higher average biomass-normalized GPP and NPP values relative to control" – was there an optimum grazing frequency for pasture regeneration?*

We now provide a comment on this topic in l. 440-442.

**Discussion**

*Management-related differences between villages could be discussed in more depth? Effects of SI between sites (expected to be stronger on the more extensive = poorer site)?*

We have added this aspect as an additional question in our list of questions in the introduction (l. 83/83), and discuss our insights on that topic in l. 475-479.

*Did more frequent grazing on certain areas affect the regeneration of pasture (positively or negatively)?*

The effect of grazing intensity/frequency on pasture regeneration was not investigated in this study. However, we focus on pasture regeneration in the context of drought and grazing in another study that is

currently in preparation (Behn et al., in prep). We have added this as a statement in the discussion (l. 524-526).

*Line 376: "SI-measures could result in yield losses in dry years due to enhanced crop growth and associated increased water demand" – yield loss due to enhanced crop growth sounds paradoxical. It is explained in lines 386 ff and I would suggest to move this sentence there.*
We have move the explanation as suggested. (l. 484-486).

*Line 385: Not sure whether N input would increase cowpea growth.*
We have added a brief discussion of the potential role of N-limitation for legumes in l. 482-484.

*Line 396: Loosening sandy soils and increasing infiltration to increase plant growth – these measures appear to be more appropriate for heavy soils.*
Soils at both villages are mixed sandy clay-loams that can be prone to slake, hard-setting and surface crusting and therefore may benefit from loosening where such problems occur. We have added this point to the discussion in l. 502-505.

*Lines 408 ff.: Could undergrazing be a problem (grass becoming moribund)? Does the model account for stimulated regrowth by grazing? Is pasture quality considered in the model? This is mentioned later (lines 454ff.), perhaps both paragraphs could be better connected.*
We have moved the corresponding section so that it is now integrated in the suggested location (l. 519 ff.)

*Line 423: Sounds unlikely that vegetation growth reacts with 2-3 months delay to the onset of rains.*
This is poorly phrased. Vegetation growth starts shortly after the onset of the rains, but peak biomass is reached with a delay of 2-3 months after the beginning of the wet season. During the time when biomass in-growth takes place, quantities a) are not yet sufficient to fully supply the demand and b) grazing can additionally slow the development. We now xplain this more clearly in the revised manuscript (l. 538ff.)

*Section 4.3: Surprising that farmers do not store crop residues; this is common practice from Senegal to Ethiopia in densely populated areas. In this context, population density in the research area might be interesting (in the MatMet section).*
We have added this information to the discussion (l. 550).

*Line 444f.: Pasture quantity is only part of the problem, correct, and so is pasture quality (high lignin contents, if moribund biomass or standing litter are fed).*
Correct, quality is also important, although it is currently only crudely accounted for in the aDGVM2 grazing scheme, where dead biomass is assigned ⅔ of the nutrition content of living grass biomass. We have added this point to the discussion (l. 564-566).

*In context with line 258, stronger effect of SI at Gabaza (one would expect higher impact in the poorer environment, i.e. in Selwana) could be discussed.*

We have added this aspect as an additional question in our list of questions in the introduction (l. 83/83), and discuss our insights on that topic in l. 475-479.

**Conclusions**

*Line 468: Holistic management is a much disputed strategy that should probably not be introduced in the last section without further explanation. What is meant here is probably integrated crop-livestock systems.*
We have rephrased this and exchanged the word "holistic" with "integrated", as suggested.

*Fig S1: As bars are the same and only the y-axis label differs, one of both subfigures could be omitted.*
We have adjusted the figure according to the suggestion and merged both panels by adding a second y-axis on the right-hand side of the first panel.

*Fig S2: Differences between solid and hatched signature are not visible*
We have changed the coloring of the hatched signature to contrast more strongly against the solid signature, and additionally changed the angles of the hatched signature to allow easier distinction of the three crop types.

*S3 and S4: Why is biomass demand for CO (no animals) > 0?*
Based on the reviewer's suggestion, both figures are now part of the main manuscript in exchange against Table 2, which has been moved to the supplementary material. As mentioned in the methods section, we prescribed a very low demand for spinup and control to establish a grass community that is generally accommodated to grazing. Even without livestock, grazing by small game is common and creates some small background demand in the CO-scenario. We have added this explanation to the captions of both figures.

*Fig.s 7 and following: What are the red and blue lines at the bottom (should be explained in the captions, so that the figures are self-explaining)? Why are standard deviations for CO sometimes (around 2007 in Gabaza A1 and A3) higher than for the grazed treatments?*
The meaning of these lines is now explained in the captions of the respective figures (it is the animal presence timelines for the two grazing scenarios).

REVIEWER 2

*The abstract would benefit from more numbers to make it more quantitative to help describe exactly what was found.*
We have added some numbers for the key results.

*There are minor usage issues throughout, the first is on line 34 (p. 2): "Livestock" is plural so the correct usage is 'Livestock provide..' (see also line 35; a quick review will correct any minor issues.) More: space after the period on L. 70, etc. (L. 89: "and cowpea" to distinguish between tubers after the comma.)*
We have included the editorial hints throughout the manuscript.

*I found the last paragraph of the Introduction to be a bit confusing: listing the questions first and then the approach used would help lead into the specific study.*

We have restructured the paragraph to improve the flow of the logic. The research questions have been moved up to precede the description of our research approach (l. 78-88).

*Please describe t/ha to distinguish between metric tons (or tonnes) and imperial tons. Obviously the former is more appropriate and I assume is used here, but the latter is in common usage in many places.*

It is metric tons. We now specify the explicit unit (metric tons) when first introducing the unit in l. 127, l. 191 and l. 322.

*LU is not defined at first use.*

Livestock unit. We have added the definition of the abbreviation at first introduction (l. 149).

*Results: I feel that there are probably too many significant digits for a modeling study throughout. For example 53 +/- 23 is probably more correct than 53.2 +/- 22.9, etc. The results were comprehensive but somewhat long, and an eye toward brevity would improve the Results section.*

We have checked the number of significant digits throughout the manuscript, and reduced them where appropriate.

*I guess that my biggest question regarding the outcomes is the suggestion for irrigation feasibility? This usually involves considerable expense and can have other deleterious consequences. A brief analysis of the likelihood or sustainability of irrigation would strengthen the conclusions.*

We have expanded this topic in our discussion (l. 493-499).